# CLAM: Class-wise Layer-wise Attribute Model for Explaining Neural Networks

## Abstract

Deep learning techniques have been actively researched for solving complex and diverse problems, demonstrating high performance across various AI domains. However, the complexity and opaqueness of deep learning-based models often make them "black boxes," leading to concerns about transparency and trustworthiness, which in turn hampers their real-world applicability. To address this issue, numerous studies have been carried out to interpret deep learning models and provide explanations for their predictions or propose interpretable new structures. In this paper, we introduce the Class-wise Layer-wise Attribute M¡model (CLAM), which aims to provide more accurate and detailed explanations for model predictions in image classification. Specifically, CLAM is designed to work in conjunction with a pre-trained image classification model and an existing interpretable algorithm to learn *class-wise layer-wise attributes* from the model features. Additionally, when generating a relevance map for new input images, CLAM leverages the learned attribute information to enhance the areas related to the target class thereby improving accuracy. Furthermore, we identify and present the influence of specific samples from the training dataset on the calculated relevance map, offering a higher level of explanation compared to existing methods. To validate the effectiveness of our proposed model, we present quantitative and qualitative experimental results using CUB-200-2011 and ImageNet datasets, along with pre-trained VGG-16 and ResNet-50 image classification models and well-known explainable models.

## 1 Introduction

Deep learning has demonstrated remarkable success in various AI tasks, including image classification, machine translation, speech recognition, and various generative models, through the development and training of specialized neural networks. Despite their efficiency, these models often function as 'black boxes' due to their intricate internal workings, which are often incomprehensible to humans, thus hindering user trust and real-world adoption due to their lack of transparency. Early approaches to mitigate this transparency issue focused on visualizing internal activations and gradients. The field has evolved chiefly in two directions: One approach generates contribution maps (or relevance maps), which use visualized internal model values to shed light on predictions. This method, however, is more about reverse-analyzing the model's decision-making rather than understanding the model itself, providing a 'visualization' rather than an 'explanation' for predictions. An alternative strategy designs inherently transparent deep learning models. While this approach enhances interpretability, it demands additional training. These models often incorporate explanatory modules or additional layers, improving performance but not necessarily providing precise 'explanations' for predictions. Both strategies have their merits and limitations, primarily offering explanations through the analysis and visualization of intermediate processes.

In this paper, we propose *CLAM* (Class-wise Layer-wise Attribute Model), a model designed to enhance the explainability of image classifiers by learning *attributes* at each class and layer level. CLAM achieves this through three primary objectives. First, it learns class-wise and layer-wise attributes by leveraging features from image classifiers and relevance maps from explainers. Attributes fundamentally extract sub-feature areas related to specific classes using features and are defined independently across various layers in the image classifier. Second, CLAM enhances the performance and noise reduction of the relevance map by reflecting information about the current

target class using the learned attributes. Lastly, CLAM identifies and presents related images from the training dataset using the attributes reflected in the relevance map. By doing so, CLAM not only offers relevance maps but also improves explainability by indicating which areas of the relevance map are influenced by specific portions of the training data.

## 2 RELATED WORKS

**Approaches to Post-hoc Explanation**: Post-hoc explanation methods, as previously described, generate relevance maps for model predictions without additional training, utilizing the internal values of the model. This approach began with the visualization of initial model activations or gradient values such as DeconvNet (Noh et al., 2015). Subsequent efforts have been made to construct relevance maps utilizing the gradient values of the models (Simonyan et al., 2013; Springenberg et al., 2014; Sundararajan et al., 2017; Kapishnikov et al., 2019; Jeon et al., 2022). Notably, as gradient values are not only essential by-products generated during the training of deep learning models but also encapsulate the variations between inputs and predictions, they continue to be extensively employed. Additionally, there are algorithms in the Layer-wise Relevance Propagation (LRP) family that slightly diverge from this approach (Bach et al., 2015; Gu et al., 2019; Iwana et al., 2019; Eberle et al., 2020; Nam et al., 2021; Naseer et al., 2021; Chefer et al., 2021). These algorithms leverage the activation and weight values of the model, performing backpropagation on a layer-wise basis to calculate contributions to predictions. This method, which essentially inverts the forward prediction process of the model regarding the input to determine pixel-level contributions, continues to be the focus of extensive research to date.

**Approaches to Transparent Design**: Unlike post-hoc explanation methods, there are approaches designed to inherently enhance model transparency by modifying the model structure itself. For instance, there have been research efforts aimed at integrating traditional machine learning methodologies with deep learning models (Angelov & Gu, 2018; Yang et al., 2018; Zhang et al., 2019; Csiszár et al., 2020). Recently, there has been a surge in research focusing on methods that incorporate specific layers like 'Attention', training the entire model, and subsequently extracting information that contributed to the model's classification. Notably, there are studies, such as ProtoPNet (Chen et al., 2019), which has significantly inspired our work. ProtoPNet not only generate relevance maps with decent model classification performance but also provide related example images by incorporating additional layers capable of internally learning prototypes. This allows for not only decent model prediction performance but also the provision of explanations (relevance maps) for predictions. Further advanced research efforts in this direction continue to emerge and evolve, as evidenced by recent studies (Nauta et al., 2021; Donnelly et al., 2022).

## 3 PROPOSED METHOD: *CLAM*

### 3.1 CLAM ARCHITECTURE

The primary goal of CLAM is to learn class-wise and layer-wise attributes from a given image classifier and explainer, and to utilize these attributes in generating relevance maps. Attributes, as previously described, correspond to sub-feature areas in a specific layer's feature that significantly influence the target class. Hence, if the attributes per class and layer are effectively learned, it's feasible to identify and reflect the areas contributing to the target class in the relevance map using the attribute information.

The left side of Figure 1 depicts the architecture of the proposed CLAM applied to a pre-trained model with multiple convolutional layers, while its right panel presents CLAM's final result. CLAM primarily consists of several CLA-Layers (Class-wise Layer-wise Attribute Layers), which can be allocated to either every or specific designated layers of the image classifier, taking as input both the connected layer's feature and the explainer's relevance map. Each CLA-layer possesses a predetermined number of attribute variables, learning sub-feature regions significant to a particular class from the classifier's features, enhancing the adjustment of the relevance map. The right panel of Figure 1 shows a relevance map for the given input image. CLAM helps to improve performance by adjusting regions in relevance map through CLA-layers. Moreover, by utilizing the learned at-

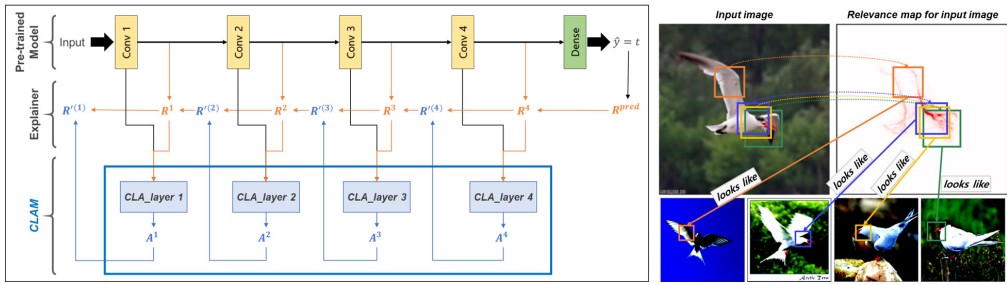

Figure 1: Overview of the CLAM Architecture and Outputs. The left side illustrates the architecture of CLAM, highlighting multiple CLA-layers that operate in conjunction with an image classifier and explainer. This synergy facilitates the generation of noise-reduced relevance maps through the utilization of learned attributes. On the right, the final output is displayed, showcasing a refined relevance map alongside images that are associated with the attributes reflected in the map.

tributes in CLA-layers, it provides examples related to key areas within each relevance map, thereby increasing its explainability. and precision of relevance maps for target classes

## 3.2 CLA-LAYER

The CLA-layer, fundamentally inspired by the prototype layer of ProtoPNet, adapts its prototype concept, using it as an *attribute* within the CLA-layer. Unlike the prototype layer—which functions as a middle layer for classification following the model's final convolutional layer—the CLA-layer is mapped one-to-one with a layer in the image classifier. The number of CLA-layers depends on the model's architecture or user configurations. An explainer linked to CLAM employs LRP family algorithms to generate a relevance map per layer, with the CLA-layer's propagation direction mirroring that of LRP, extending from the classifier's output to input layer.

Figure 2 illustrates the propagation step within a single CLA-layer. For the $l^{th}$ CLA-layer, it takes in feature $\boldsymbol{F}$ ($\mathbf{W}^{(l)} \times \mathbf{H}^{(l)} \times \mathbf{D}^{(l)}$) from the corresponding classifier layer. A sub-feature $\boldsymbol{S}$ ($\mathbf{P}^{(l)} \times \mathbf{P}^{(l)} \times \mathbf{D}^{(l)}$) is then extracted based on a predetermined patch size, normalized, and passed through the 'Add-on-Layer (AoL)' block (comprising two $1 \times 1$ convolutional layers), yielding the output $\boldsymbol{S'}$. The normalization of $\boldsymbol{S}$ is crucial due to the layer-wise training algorithm of CLAM, explained in detail in subsequent sections. In CLAM's architecture, each CLA-layer is trained sequentially rather than simultaneously to prevent the 'feature collapse' issue, which occurs when a pre-trained layer's outcomes hinder the subsequent layer from developing new feature representations. To mitigate this, the output feature vector from one CLA-layer is normalized before being input into the next, maintaining only relative magnitude during learning by considering vector length information. The AoL block plays a preprocessing, mapping sub-features $\boldsymbol{S}$ into class-specific spaces within each layer throughout the learning phase.

Subsequently, within the CLA-layer, the $L^2$ distance between attribute $\boldsymbol{A}$ and $\boldsymbol{S'}$ is computed to derive the distance map $\boldsymbol{D}$. The attribute $\boldsymbol{A}$ is characterized by its dimensions ($\mathbf{N}^{(l)} \times \mathbf{A}^{(l)} \times \mathbf{A}^{(l)} \times \mathbf{D}^{(l)}$), where $\mathbf{N}^{(l)}$ denotes the number of attributes in $l^{th}$ CLA-layer, $\mathbf{A}^{(l)}$ signifies the dimension of each attribute. The size of $\mathbf{A}^{(l)}$ is determined based on the predefined patch size within each CLA-layer, satisfying the condition $\mathbf{P}^{(l)} > \mathbf{A}^{(l)}$. The values in $\boldsymbol{D}$ represent the distance between sub-features and respective attributes. The distance value decreases as the similarity between attributes and sub-features increases. During the training phase of the CLA-layer, the loss is computed with an objective to minimize the values in $\boldsymbol{D}$ (details will be discussed in Section 3.3). After training, the regions in the sub-feature space exhibiting the smallest $\boldsymbol{D}$ values are identified and integrated into the explainer's relevance map to refine it.

Within the CLA-layer, the $L^2$ distance between attribute $\boldsymbol{A}$ and $\boldsymbol{S'}$ is calculated to produce distance map $\boldsymbol{D}$. Attribute $\boldsymbol{A}$ possesses dimensions ($\mathbf{N}^{(l)} \times \mathbf{A}^{(l)} \times \mathbf{A}^{(l)} \times \mathbf{D}^{(l)}$), with $\mathbf{N}^{(l)}$ and $\mathbf{A}^{(l)}$ representing the number and dimension of attributes in the $l^{th}$ CLA-layer, respectively. The size of $\mathbf{A}^{(l)}$ is

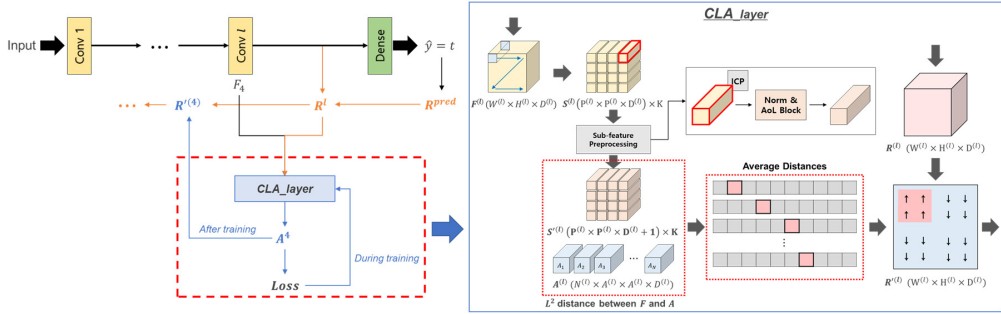

Figure 2: The figure's left panel displays a CLA-layer connected to an image classifier layer, with the right panel detailing the CLA-layer's structure and process.

predetermined by the patch size within each CLA-layer, adhering to the condition $\mathbf{P}^{(l)} > \mathbf{A}^{(l)}$. The values in $\boldsymbol{D}$ denote distances between sub-features and attributes, decreasing as their similarity increases. During the CLA-layer's training phase, the loss, aiming to minimize $\boldsymbol{D}$ values, is computed (discussed in detail in Section 3.3). Post-training, regions in the sub-feature space with minimal $\boldsymbol{D}$ values are identified and incorporated into the refined explainer's relevance map.

### 3.3 TRAINING PROCEDURE

When using CLAM to adjust relevance maps, inaccuracies in extracted sub-feature regions from input features may occur prior to attribute training, potentially leading to progressively inaccurate relevance maps during propagation through CLA-layers. To mitigate this, CLAM incorporates the Forward-Forward (FF) algorithm (Hinton, 2022), which employs two forward passes instead of the traditional forward and backward passes in backpropagation. This process involves forwarding positive data with actual labels and negative data with random labels, effectively transforming the task into binary classification. This approach adjusts weights to modify the 'goodness' of the results propagated through the hidden layer, with training proceeding layer-by-layer. For effective FF algorithm implementation in CLAM, two considerations are crucial: generating positive and negative data samples from the original image dataset and defining 'goodness' as a loss function for weight adjustment within the CLA-layer. The generation of positive and negative data necessitates the creation of Intrinsic Class Pattern (ICP) introduced in SymBa (Lee & Song, 2023), offering a unique two-dimensional matrix pattern for each class.

After generating the ICP, we can generate the positive and negative sub-features $\boldsymbol{S}$, presumed relevant to the target class. Since there isn't a clear ground-truth label available for training $\boldsymbol{S}$, we utilize the relevance map generated by the explainer. Equation 1 outlines the extraction of sub-feature $\boldsymbol{S}$ by identifying coordinates with high relevance values from relevance map $\boldsymbol{R}$ and retrieving the corresponding sub-features from $\boldsymbol{F}$ in $l^{th}$ layer.

$$(x_i, y_i) = \max(AvgPool(\boldsymbol{R}_{2d}^{(l)})), \ i = \{1, \ldots, N\}$$
$$\boldsymbol{S}_{(p,i)}^{(l)} = \boldsymbol{F}_{x_i:x_{i+p}, y_j:y_{j+p}, :}^{(l)} + \boldsymbol{C}_y, \ \boldsymbol{S}_{(n,i)}^{(l)} = \boldsymbol{F}_{x_i:x_{i+p}, y_j:y_{j+p}, :}^{(l)} + \boldsymbol{C}_{r \neq y} \tag{1}$$

In Equation 1, $\boldsymbol{R}^{(l)}2d$ is a 2D map obtained by selecting the maximum value per pixel across all channels from $\boldsymbol{R}^{(l)}$, the relevance map of the $l^{th}$ layer. This 2D map is then processed through an AvgPooling layer with a kernel size corresponding to the CLA-layer patch size. The N coordinates $((x_i, y_i))$, representing points significantly contributing to the target class in the current layer, are extracted from $\boldsymbol{R}^{(l)}2d$ by applying top-k function (k=$N$). Using these coordinates, sub-features of patch size are extracted from $\boldsymbol{F}^{(l)}$. For training based on FF algorithm, ICPs $\boldsymbol{C}_y$ and $\boldsymbol{C}_{r \neq y}$ for target class $y$ and random class $r$ are appended to $\boldsymbol{S}$'s final channel for positive and negative sub-features $\boldsymbol{S}^{(l)}(p,i)$ and $\boldsymbol{S}^{(l)}(n,i)$ for the $i^{th}$ attribute. After generating $\boldsymbol{S}^{(l)}(p,i)$ and $\boldsymbol{S}^{(l)}(n,i)$, each undergoes forward propagation within the CLA-layer, with normalization and the AoL block applied as outlined in Section 3.2, yielding $\boldsymbol{S'}^{(l)}(p,i)$ and $\boldsymbol{S'}^{(l)}(n,i)$ as detailed in Equation 2.

$$\boldsymbol{S}_{(p,i)}'^{(l)} = \text{AoL}(||\boldsymbol{S}_{(p,i)}^{(l)}||), \ \boldsymbol{S}_{(n,i)}'^{(l)} = \text{AoL}(||\boldsymbol{S}_{(n,i)}^{(l)}||) \tag{2}$$

As outlined in Equation 3, each attribute in the CLA-layer calculates the $L^2$ distance relative to its corresponding $\boldsymbol{S}'^{(l)}(p,i)$ and $\boldsymbol{S}'^{(l)}(n,i)$.

$$\boldsymbol{D}_{(p,i)}^{(l)}, = L^2 - \text{Distance}(\boldsymbol{S}_{(p,i)}'^{(l)}, \boldsymbol{A}_i^{(l)}), \ \ \boldsymbol{D}_{(n,i)}^{(l)} = L^2 - \text{Distance}(\boldsymbol{S}_{(n,i)}'^{(l)}, \boldsymbol{A}_i^{(l)}) \tag{3}$$

$\boldsymbol{A}^{(l)}i$ represents each attribute in the $l^{th}$ layer, with $\boldsymbol{D}^{(l)}(p,i)$ and $\boldsymbol{D}_{(n,i)}^{(l)}$ denoting the distance matrices between the sub-feature and its respective attribute. In adherence to the FF algorithm, the training goal for attributes is twofold: minimizing $\boldsymbol{D}^{(l)}(p,i)$ from the positive pass while maximizing $\boldsymbol{D}^{(l)}(n,i)$ from the negative pass. This is facilitated using the Balanced Contrastive Loss (BCL) introduced in SymBa, with Equation 4 delineating the adjusted BCL loss for CLA-layer training.

$$\Delta = \beta \max(\frac{1}{NWH}\sum_i\sum_{w,h}\boldsymbol{D}_{(p,i),w,h,:}^{(l)}) - (1-\beta)\,(\frac{1}{WHN^2}\sum_i\sum_{w,h}\boldsymbol{D}_{(n,i),w,h,:}^{(l)}) \tag{4}$$

$$L = log(1+e^{\alpha\Delta}); \ \boldsymbol{D}_p^{(l)} \to 0, \ \boldsymbol{D}_n^{(l)} \to \infty \Leftrightarrow L \to 0 \tag{5}$$

In this formulation, $N$, $W$, and $D$ denote the number of attributes and the attribute's width and height, respectively. $\Delta$ calculates the per-channel average distance for $\boldsymbol{D}^{(l)}(p,i)$ and $\boldsymbol{D}^{(l)}(n,i)$, normalizing the distance magnitude by dividing by the attribute count. For the positive distance matrix $\boldsymbol{D}^{(l)}(p,i)$, the maximum value is extracted (since the objective is minimization), while for the negative matrix $\boldsymbol{D}^{(l)}(n,i)$, an average value is computed across all attribute outcomes. $\beta$ regulates the balance between positive and negative outcomes, with the loss computed based on the derived $\Delta$ value. The *Softplus* function is utilized as the loss function, with $\alpha$ modulating the magnitude of $\Delta$. This loss minimization process fine-tunes the CLA-layer weights, aiming to decrease $\boldsymbol{D}^{(l)}(p,i)$ and increase $\boldsymbol{D}^{(l)}(n,i)$. Upon the completion of training for a single CLA-layer, the learned attribute for a given feature locates the sub-feature region where the average value of $\boldsymbol{D}_{(p,i)}^{(l)}$ is minimized. Then, the values within the corresponding areas of the relevance map are increased, thereby amplifying the contribution of regions associated with the current target class.

### 3.4 Generating Explanation

Upon the completion of training across all CLA-layers, CLAM, in tandem with the explainer, yields the refined relevance map and associated images, shedding light on the model's predictions. Each CLA-layer discerns attribute regions—sub-features pivotal to the target class. During the testing phase, attributes evaluate all sub-features from the input feature with patch size, selecting those with minimal average distance values. Consequently, a mask is generated and applied to the relevance map to enhance values in selected regions, highlighting target class-relevant areas. Initialized with user-defined strength values (within a 1.1 to 1.3 range in our experiments), this mask is utilized to adjust the relevance map, which is then propagated to subsequent layers for recursive processing until the input layer is reached. Furthermore, attributes within each CLA-layer create $\boldsymbol{D}^{(l)}(p,i)$ for both the test image and all training set images, calculating the similarity between $\boldsymbol{D}^{(l)}(p,i)$ of the test and training images. By identifying training images with the highest similarities, CLAM efficiently offers images that are related to the relevance map regions at each layer as previously illustrated in Figure 1.

## 4 Experimental Evaluations

### 4.1 Experimental Setup

In the conducted experiments, we use notable CNN-based image classifiers, namely VGG-16 and ResNet-50, with datasets CUB-200-2011 (Wah et al., 2011) and ImageNet (Russakovsky et al., 2015), which are prevalent in image classification research. We integrate CLAM with explainers utilizing LRP-based algorithms, like CLRP and SGLRP, for comparison with various esteemed explainers, including gradient-based methods and CAM. Our experimental evaluation consists of qualitative and quantitative analyses. The former involves displaying and analyzing relevance maps on sample images and related training image examples, elucidating the relevance map's connection to the target class. The latter includes a comparison and analysis of relevance maps produced by different explainers and CLAM using multiple evaluation metrics.

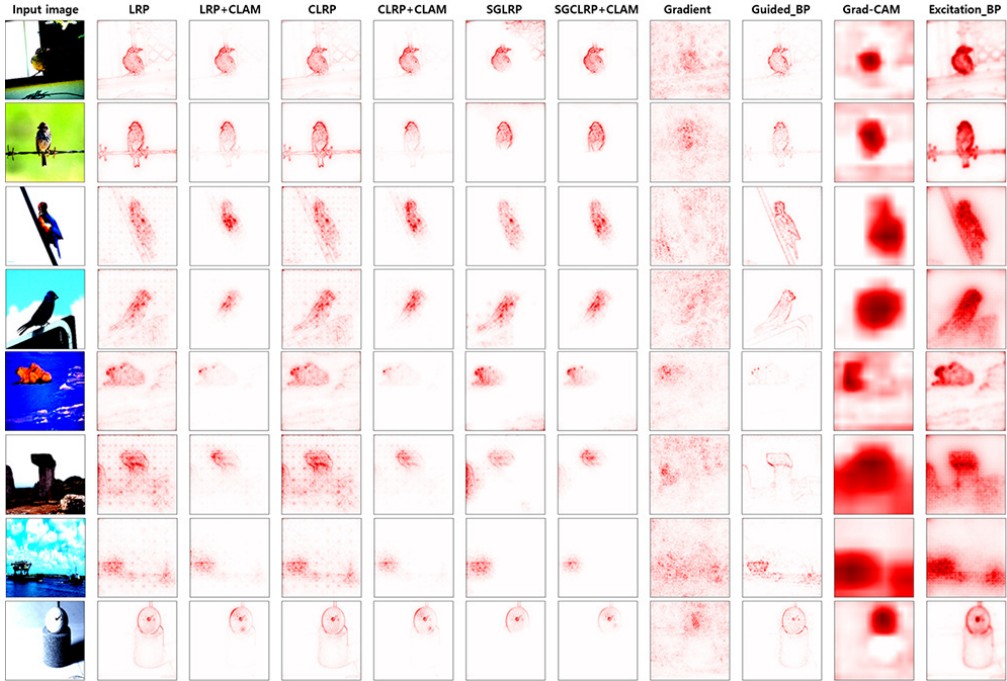

Figure 3: Comparison of generated relevance maps. Columns present relevance maps from different explainers for the input image, with results for CUB images shown in Rows 1-4 and ImageNet images displayed from Row 5 onward.

## 4.2 QUALITATIVE EVALUATION

CLAM aims to improve generated relevance performance and offer additional explanations by learning layer-wise attributes by serving a supplementary role by leveraging class-wise, layer-wise attributions. To validate its effectiveness, we initially conduct a qualitative assessment comparing relevance maps produced by traditional LRP-based algorithm such as LRP and CLRP, those integrated with CLAM and other explainers such as CAM and Guided_Backprop. We also conduct a quantitative evaluation with benchmarks for various algorithms

### 4.2.1 COMPARATIVE ANALYSIS OF RELEVANCE MAPS GENERATED BY VARIOUS METHODS

Figure 3 displays relevance maps generated by various explainers for a given input. Specifically, rows one through four utilize validation image samples from the CUB dataset as input. For these inputs, the first and second rows present relevance maps of the explainers for VGG-16, while the third and fourth rows depict the results for relevance maps generated for ResNet-50. When comparing the LRP-based algorithms with CLAM, it is evident that CLAM with LRP algorithm substantially reduces noise in areas outside of the object. For instance, surrounding elements like trees or structures around a bird are eliminated, highlighting the bird more prominently. In the case of relevance maps based on ResNet-50, the maps contain considerable noise since features from BatchNorm layers are also utilized during their generation. CLAM effectively mitigates such noise also.

Figure 3 shows relevance maps generated by various explainers for given inputs. Rows 1-4 illustrate relevance maps for CUB dataset validation images, with rows one and two displaying maps for VGG-16 and rows three and four for ResNet-50. CLAM paired with LRP algorithms noticeably diminishes noise outside the object area compared to standalone LRP-based algorithms, effectively isolating and highlighting the primary subject, like a bird, from its surroundings. In relevance maps derived from ResNet-50, significant noise is observed due to the incorporation of features from BatchNorm layers; however, CLAM effectively reduces this noise. Comparison with other explainers reveals that Guided_Backprop primarily yields edge-detection-like results without discerning objects. Grad-CAM highlights principal regions but lacks pixel-level precision due to its reliance

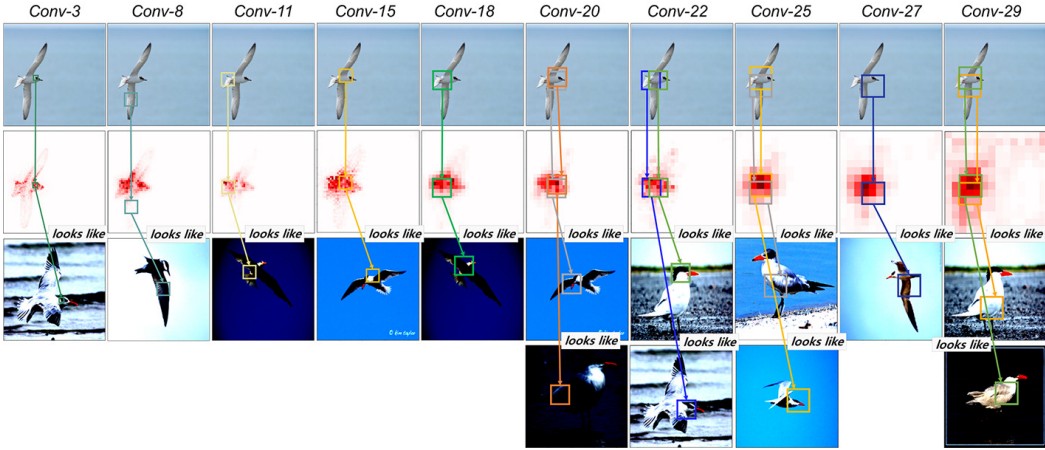

Figure 4: Examples of relevance maps alongside associated training images with bounding boxes extracted by learned attributes in each CLA-layer. All outputs are generated through CLRP+CLAM for VGG-16 pre-trained on the CUB dataset.

on resizing relevance maps from each model's final layer features. Although Excitation_Backprop generates fairly accurate relevance maps, it introduces some noise. Similar observations are made for ImageNet image samples' relevance maps. Applying CLAM to LRP-based algorithms effectively filters out areas irrelevant to the target, enhancing the precision in identifying pertinent regions while omitting unrelated ones, as evidenced in examples labeled 'megalith' (6th image) and 'drilling_platform' (7th image). In the 'golf_ball' example in last, CLAM adeptly excludes irrelevant objects beneath the ball, providing a map aligned with the assigned label.

### 4.2.2 ANALYSIS OF EXAMPLE-BASED EXPLANATIONS THROUGH ATTRIBUTES

Figure 4 demonstrates the generation of relevance maps via CLRP and CLAM for a VGG-16 model trained on the CUB dataset, with each column titled by the VGG-16 layer associated with a CLA-layer. The first row highlights areas on the input image that attributes, learned at respective CLA-layers, deem relevant to the target class. The subsequent row presents relevance maps at each layer, produced by enhancing values in attribute-selected areas via masking. Beneath each relevance map, images from the training set, closely related to areas bounded and identified by attributes, are displayed. Results of attributes pointing to identical or similar image regions are treated as redundant and are accordingly eliminated. These areas in both the input and training images are projected through attributes into a shared space. The highly similar areas, determined by attribute-created distance maps, are then visualized, allowing interpretation of relevance map-marked areas as influenced by corresponding areas in the related training images.

Taking the 'Conv-8' layer as an example, attributes emphasize the bird's wing area, which subsequently enhances the wing's visibility in the 'Conv-3' relevance map. A consistent observation across layers is that attributes primarily focus on the bird's body. The linked images from the training set also highlight the bird's body and face. This suggests that the relevance map identifies influential regions for predictions, shedding light on major areas in the relevance map impacted by specific images or regions from the training set. Further examples of attribute visualization are available in the Appendix A.4.

### 4.3 QUANTITATIVE EVALUATION

For quantitative analysis, we utilized three established evaluation techniques: Pointing Game, Perturbating LeRF, and average IoU for segmentation masks, comparing CLAM against various explainers. CLAM was benchmarked against three prominent LRP algorithms (LRP, CLRP, SGLRP), each employing LRP$-_{\alpha\beta}$ with $\alpha_1\beta_0$. Furthermore, the comparative analysis involved other saliency

Table 1: Pointing accuracy results of explainers based on VGG-16 and ResNet-50 re-trained on CUB-200-2011 and ImageNet2012. Underline results denote improvement by CLAM over the respective LRP algorithm, while **boldface** highlights the top performance for each dataset and classification model combination. 'L.A.', 'Deconv.', 'G-BP.', and 'Ex-BP.' abbreviate local approximation method, DeconvNet, Guided_Backprop, and Excitation_Backprop respectively, while '+*C*' denotes instances where CLAM is incorporated.

|  |  | LRP | LRP+*C* | CLRP | CLRP+*C* | SGLRP | SGLRP+*C* |
|---|---|---|---|---|---|---|---|
| CUB | VGG-16 | .5550 | .5651 | .5590 | **.5721** | .4835 | .4877 |
|  | ResNet-50 | .4972 | .4893 | .5350 | .5425 | .5582 | .5443 |
| ImageNet | VGG-16 | .5774 | .5850 | .5814 | **.5920** | .5355 | .5373 |
|  | ResNet-50 | .5893 | .5734 | .5892 | .5767 | **.6233** | .6212 |
|  |  | LA. | Deconv. | Gradient | G-BP. | Grad-CAM | Ex-BP. |
| CUB | VGG-16 | .3881 | .4151 | .4915 | .5385 | .5048 | .5539 |
|  | ResNet-50 | .3619 | .4525 | .4125 | .5158 | .5740 | **.5782** |
| ImageNet | VGG-16 | .4508 | .4750 | .5125 | .5449 | .5471 | .5608 |
|  | ResNet-50 | .4467 | .5081 | .5023 | .5417 | .5962 | .5943 |

map generation methods, including DeconvNet, Gradient, Linear approximation, Guided_Backprop, Grad-CAM, and Excitation_Backprop.

### 4.3.1 EVALUATION USING THE POINTING GAME

The Pointing Game method assesses explainers' localization skills by checking the alignment between high relevance values in generated maps and object bounding boxes in datasets. Relevance values above a set threshold are extracted, with those in the primary object region labeled as 'hits' and the rest as 'misses'. The pointing accuracy is then calculated as the ratio of 'hits' to the total count of 'hits' and 'misses'. An extended version of the Pointing Game is used in this experiment, as described in (Iwana et al., 2019), where values from relevance maps are filtered based on various thresholds to calculate accuracy. Thresholds correspond to different energy percentages, with higher thresholds excluding lower relevance values. In our experiment, performance is evaluated at thresholds from 0.05 to 1.0, in increments of 0.05.

Table 1 shows the Pointing Game performance comparison among various LRP algorithms, CLAM, and other explainers, with each value reflecting the average pointing accuracy across thresholds. The comparison between LRP algorithms and their CLAM-enhanced counterparts reveals performance enhancement across all datasets when CLAM is applied to LRP algorithms based on VGG-16. As evidenced in Figure 3, this improvement is likely due to CLAM's ability to mitigate noise surrounding objects in the generation of relevance maps. Performance of CLAM-enhanced LRPs based on ResNet-50 generally shows similar or slightly decreased performance compared to originals. This decrease is attributed to ResNet-50's complex structure, necessitating numerous CLA-layers, which might over-adjust relevance maps and emphasize smaller, potentially less significant object areas. Issues associated with the small final feature size of ResNet-50 and the need for a $4 \times 4$ patch size for the initial CLA-layer further complicate attribute learning, especially considering the large number of ImageNet classes. This acknowledged limitation of CLAM is further discussed in the Appendix A.2 and Appendix A.6, along with potential remedies. Despite these challenges, SGLRP demonstrates commendable performance on ImageNet when paired with ResNet-50, owing to its effective noise reduction capabilities, as illustrated in Figure 3. While CLPR+CLAM performs best with VGG-16-based relevance maps, algorithms like Excitation_Backprop and SGLRP show superior results with ResNet for the reasons outlined above.

### 4.3.2 EVALUATION THROUGH PERTURBING LERF

In this experiment, image classification performance is re-evaluated using perturbed images created by removing pixels associated with low relevance values from the original images, as indicated by each explainer's generated relevance map. Classification performance of the perturbed images is

Table 2: Results on last and mean accuracies of image classifiers for perturbed inputs. Underline indicate improved performance due to CLAM, while **boldface** denotes the best performance for each dataset and model; values closer to 1.0 are preferable.

|  | CUB-200-2011 | | | | ImageNet 2012 | | | |
|  | VGG-16 | | ResNet-50 | | VGG-16 | | ResNet-50 | |
|  | last | mean | last | mean | last | mean | last | mean |
|---|---|---|---|---|---|---|---|---|
| LRP | .7856 | .8530 | .6499 | .7270 | .6681 | .8126 | .7304 | .8542 |
| LRP+CLAM | .8004 | .8654 | .7135 | .7786 | .6831 | .8219 | .7647 | .8747 |
| CLRP | .7858 | .8530 | .6799 | .7690 | .6675 | .8126 | .7314 | .8539 |
| CLRP+CLAM | **.8015** | **.8656** | .7658 | .8351 | **.6840** | .8226 | .7693 | .8751 |
| SGLRP | .6816 | .7712 | .7627 | .8116 | .6443 | .7835 | .8547 | .8996 |
| SGLRP+CLAM | .6793 | .7534 | .7772 | .8206 | .6492 | .7817 | **.8542** | **.9034** |
| Linear Appr. | .0563 | .1171 | .0445 | .1712 | .0613 | .2210 | .2675 | .0775 |
| DeconvNet | .1985 | .4040 | .2506 | .5004 | .1572 | .4012 | .2913 | .4988 |
| Gradient | .5385 | .6927 | .2937 | .5140 | .3215 | .5591 | .3310 | .5192 |
| Guided-BP. | .6471 | .7415 | .6171 | .7180 | .5022 | .6723 | .5366 | .6232 |
| Grad-CAM | .7421 | .8303 | **.8358** | **.8878** | .6738 | **.8367** | .7158 | .7385 |
| Excitation-BP. | .7664 | .8536 | .8030 | .8640 | .6629 | .8212 | .6887 | .7222 |

then assessed; a significant decline in performance suggests the relevance map has incorrectly assigned low relevance to crucial object areas, while stable or minimally declined performance implies that low-relevance areas are accurately assigned, demonstrating an effective relevance map and explainer. This process is repeated for 100 perturbations, with 200 pixels removed in each instance, to analyze performance variation. Table 2 presents classification performance for each explainer with perturbed inputs, using images from each dataset's test set that were initially correctly classified. Performance close to 1.0 denotes stable performance with perturbations, signifying superior relevance map generation by the explainer. The 'last' and 'mean' columns represent the performance after deleting the last 20,000 pixels and the average performance across all iterations, respectively.

Comparing LRP and CLAM results, the inclusion of CLAM often enhances performance. The improvement observed with CLAM applied to ResNet-50, differing from the Pointing Game results, can be attributed to the evaluation method differences. While the Pointing Game boosts hit rates with more pixels within a set bounding box, using CLAM with SGLRP might unintentionally eliminate essential object areas, leading to performance declines. However, in perturbed image evaluations, removing some object areas does not directly lead to performance drops but can improve performance by reducing noise. When comparing CLRP+CLAM to other explainer algorithms, it generally exhibits superior performance for VGG-16. For ResNet-50, as seen in prior Pointing Game results, other algorithms slightly outperform both LRP and CLAM. Notably, SGLRP maintains commendable performance on ResNet-50 trained with ImageNet, aligning with previous observations.

## 5 CONCLUSION

In this paper, we proposed the CLAM model, which not only enhances the performance of existing layer-wise relevance map generation algorithms but also provides additional explanations for significant areas within the relevance map. Specifically, attributes learned within the CLA-layers of CLAM are capable of extracting areas relevant to given features on a per-class and per-layer basis, thereby improving the explanatory power of classification models. Analysis results from various experiments shared in the experimental section validate the efficacy of the proposed model. Furthermore, given that CLAM is in its nascent stage, there are several identified limitations. These constraints, along with future developmental plans for addressing them, are comprehensively discussed in the Appendix A.6.

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

# A  APPENDIX

## A.1  IMPLEMENTATION DETAILS

In this section, we provide implementation details including parameter settings used for experiments, preprocessing steps, the structure of the image classifier, and so on. Additionally, we present further experimental results and analyses, along with limitations and future directions. To obtain image features, two image classification models, VGG-16 (Simonyan & Zisserman, 2014) and ResNet-50 (He et al., 2016), were utilized with pre-trained weights on ImageNet dataset. In the context of the CUB-200-2011 dataset, fine-tuning was performed only on the fully connected layer of each image classifier with the number of classes of the dataset. The fine-tuned VGG-16 and ResNet-50 models achieved performance levels of approximately 68% and 66% on CUB-200-2011 dataset, respectively. We experimented with these two models across all explainers and CLAM.

In CLAM, there are several tunable parameters at the implementation level. One such parameter is '*nb_attrs*', which is provided as a list. Each element of this list specifies the number of attributes defined in each CLA-layer and the number of attributes is determined by considering both the feature size of the connected image classification layer and the attribute size in each respective CLA-layer. The length of this list corresponds to the number of classifier layers connected to the CLA-layers. In our experiments, this parameter was set in two distinct types for each image classifier. For the VGG-16, two configurations were considered: one where CLA-layers were connected to all the convolutional layers, and another where they were linked only to the convolutional layers immediately preceding each MaxPooling layer. For the ResNet-50, configurations were distinguished based on whether or not CLA-layers were connected to the intermediate convolutional layers within each bottleneck block.

For each CLA-layer, there are parameters for the patch size, '*sz_patches*,' used to extract sub-features from the features received from the connected image classification layer, as well as the size of each attribute, denoted as '*sz_attrs*'. It should be noted that the size of each attribute must always be smaller than the patch size. As previously described, a mask is generated within each CLA-layer to reflect the key regions extracted by attributes onto the relevance map. The mask has the same dimensions as the patch size of the CLA-layer and is applied to the coordinates where the $l^2$-distance between attributes and sub-features in the relevance map is minimized. In practice, the application of the mask involves multiplying it with the relevance values within the extracted. The center value of the mask is initialized by '*strength*' parameter and neighboring values are set to have higher values for CLA-layers closer to the prediction layer (ranging from 1.2 to 1.4) and decreases for those farther away (ranging from 1.1 to 1.15).

The parameters used for training and loss function in each CLA-layer are '*alpha*' and '*beta*' described in the main text. *alpha* values are set within the range of 0.01 to 0.1 depending on CLA-layer, while *beta* values are set between 0.5 and 0.6. The learning rate is initially set at $0.01 \times \left(\frac{sz\_attrs_i}{2}\right)$ and is scheduled to decrease by 95% with each epoch during the training of individual CLA-layers. The maximum number of epochs for each layer is set to 100, and early stopping is applied; training ceases if the average minimum distance value for the positive data does not improve beyond a certain number of epochs. For software and experimental platform, all experiments were conducted on two Linux machines with the following configurations: (1) NVIDIA GTX 3090TI, Intel Core i9-9900K CPU, 64GB Memory, and (2) NVIDIA A100, Intel Xeon Gold 6230R CPU, 125GB Memory. All implemented code and techniques used in the experiments were tested on the Python-based PyTorch deep learning framework (v1.12.1).

## A.2  REVISITING THE POINTING GAME AND CONDUCTING A DETAILED ANALYSIS OF RESULTS

In Section 4.3.1, we have previously compared the average pointing accuracy between CLAM and other explainers through the extended Pointing Game. Upon this comparison, CLAM exhibited improved performance over the traditional LRP algorithms for VGG-16. For a more in-depth analysis of these findings, Figure 5 presents graphs of pointing accuracy for each threshold value. Each graph corresponds to VGG-16 and ResNet-50, which were pre-trained on CUB-200-2011 and ImageNet, respectively. The dotted lines in the graphs represent the original LRP algorithms (green: LRP, blue: CLRP, red: SGLRP), while the solid lines depict the performance when each LRP algorithm is

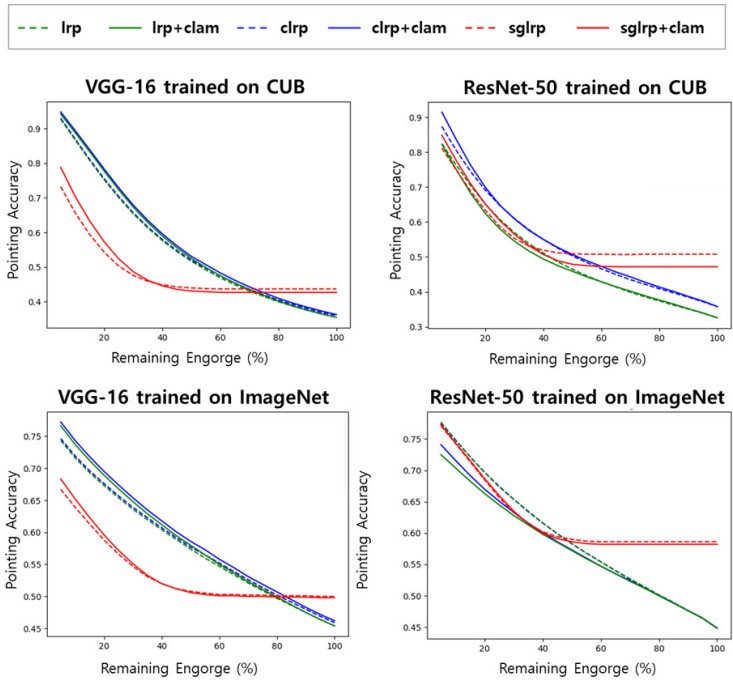

Figure 5: Comparison of LRP Algorithm and Combined Performance of LRP+CLAM Across Datasets and Image Classifiers. Each graph represents pointing accuracy corresponding to residual energy levels, where dotted lines illustrate original LRP algorithms such as LRP, CLRP, and SGLRP, and solid lines denote combinations of each LRP algorithm with CLAM.

combined with CLAM. The x-axis of the graphs indicates the residual energy levels, and the y-axis represents the pointing accuracy corresponding to those energy levels. -Energy($E$) is related to the threshold as below.

$$E = \frac{\#(R_t^{(l)} >= \tau)}{\#(R_t^{(l)} > 0)} \tag{6}$$

In the aforementioned formula, $\tau$ represents the threshold value. Consequently, as the threshold value increases, relevance values in the given relevance map that fall below the threshold are not utilized, leading to a reduction in residual energy levels. If $E$ is 100%, the threshold becomes zero, implying that all relevance values are incorporated in the computation of pointing accuracy.

Observing the graphs in the figure, it is evident that LRP and CLRP exhibit similar patterns across all residual energy levels, with CLRP marginally outperforming LRP overall, and the application of CLAM results in slight performance improvement. This aligns with our previous analysis of average pointing accuracy. For SGLRP, it displays lower performance than both LRP and CLRP at higher residual energy levels; however, its performance improves as the residual energy increases beyond a certain point. This improvement suggests that SGLRP benefits from utilizing more relevance maps and is more adept at handling noise than its counterparts. Consequently, we infer that SGLRP inherently possesses the capability to effectively eliminate noise during the generation of relevance maps. In this context, applying CLAM to SGLRP does not necessarily result in performance improvement compared to when it is applied to LRP or CLRP. CLAM inherently reduces noise in regions unrelated to the target class by identifying areas related to attributes learned at each CLA-layer and reflecting these in the relevance map. Therefore, applying CLAM to SGLRP, which already excels in noise reduction, might not substantially boost its effectiveness or might even inadvertently eliminate areas of the object, leading to counterproductive outcomes, as can be observed in Figure 7.

Figure 6 presents the pointing accuracy per residual energy level for various explainers including LRP, CLAM, and others, across different datasets and image classifiers. Each column corresponds

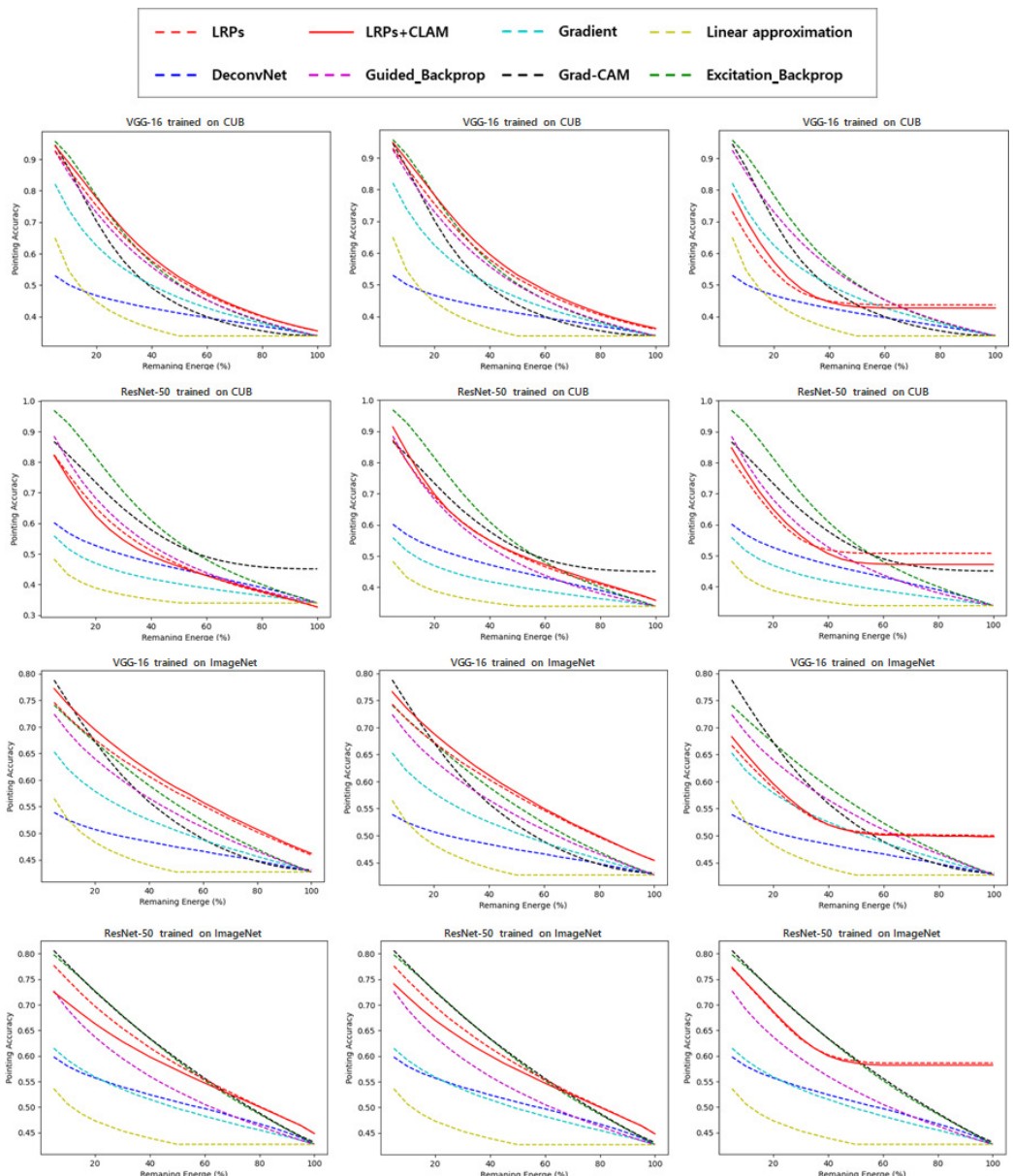

Figure 6: Comparison of pointing accuracy performance between the LRP algorithm and other explainers across various datasets and image classifiers.

to LRP & LRP+CLAM, CLRP & CLRP+CLAM, and SGLRP & SGLRP+CLAM, with dotted lines representing the original explainers including LRP, and solid lines indicating the performance of the respective LRP+CLAM combination. For the VGG-16 model, LRP/CLRP+CLAM generally outperforms other algorithms across most residual energy levels, particularly when all relevance values are utilized, suggesting a lower level of noise in the original relevance map and highlighting the noise reduction efficacy of CLAM. For ResNet-50, the performance varies, with either CLAM or other algorithms taking the lead depending on the situation. The observed performance variations for LRP and CLAM on ResNet-50 are related to its architecture, which will be further discussed in Appendix A.6 to solve this issue.. Performance improvement is noted when solely comparing LRP and CLAM on ResNet-50. Observing rows 1 and 2 of column 4 in Figure 6, performance is better without CLAM at high residual energy levels, but as this level increases, the performance with CLAM improves. This implies that while filtering the relevance map by threshold with CLAM may

Table 3: Results on lower and higher pointing accuracies of explainers based on VGG-16 and Resnet-50 pre-trained on CUB-200-2011 and ImageNet2012 datasets. Underline indicate improved performance due to CLAM, while **boldface** denotes the best performance for each dataset and model; values closer to 1.0 are preferable.

| | CUB-200-2011 | | | | ImageNet 2012 | | | |
| | VGG-16 | | ResNet-50 | | VGG-16 | | ResNet-50 | |
| | Lower | Higher | Lower | Higher | Lower | Higher | Lower | Higher |
|---|---|---|---|---|---|---|---|---|
| LRP | .6962 | .4138 | .6116 | .3827 | .6501 | .5048 | .6707 | .5080 |
| LRP+CLAM | .7137 | .4165 | .5946 | .3839 | .6636 | .5065 | .6616 | .5052 |
| CLRP | .6992 | .4188 | .6522 | .4177 | .6531 | .5098 | .6709 | .5079 |
| CLRP+CLAM | **.7197** | .4245 | .6625 | .4225 | **.6696** | **.5145** | .6779 | .5055 |
| SGLRP | .5296 | **.4373** | .6089 | **.5075** | .5691 | .5019 | .6603 | **.5863** |
| SGLRP+CLAM | .5483 | .4271 | .6159 | .4726 | .5748 | .4998 | .6607 | .5817 |
| Linear Appr. | .4371 | .3391 | .3844 | .3394 | .4750 | .4266 | .4664 | .4270 |
| DeconvNet | .4575 | .3729 | .5119 | .3931 | .5004 | .4496 | .5472 | .4691 |
| Gradient | .5958 | .3872 | .4583 | .3666 | .5645 | .4605 | .5464 | .4582 |
| Guided-BP. | .6784 | .3985 | .6404 | .3911 | .6173 | .4726 | .6140 | .4695 |
| Grad-CAM | .6431 | .3666 | .5809 | .4670 | .6367 | .4575 | **.6941** | .4984 |
| Excitation-BP. | .7102 | .3977 | **.7419** | .4144 | .6429 | .4787 | .6928 | .4958 |

reduce performance due to excessive noise removal, using all relevance values can enhance performance through noise reduction by CLAM. For SGLRP, its performance pattern differs from other LRP algorithms, demonstrating higher performance as residual energy levels increase, consistent with the observations from Figure 5. This phenomenon is especially pronounced with ResNet-50.

Table 4 displays the average pointing accuracy of various explainers for both Lower and Higher cases. The Lower and Higher cases represent the average pointing accuracies below and above a threshold of 0.5, respectively. Examining the performance of LRP and CLAM, the beneficial effect of CLAM, as noted in previous results, is evident in many instances. Observing the Lower case, specifically for VGG-16, CLRP+CLAM outperforms other algorithms, suggesting that filtering relevance values by threshold followed by the application of CLAM allows for additional noise reduction. For ResNet-50, Excitation-Backprop and Grad-CAM respectively exhibited commendable performance on the CUB-200-2011 and ImageNet datasets. Furthermore, in the Higher case, SGLRP frequently showed superior performance, implying that the algorithm inherently minimizes noise effectively.

## A.3 Additional Relevance Map Examples And Analysis

Figures 7 and 8 display the results of relevance maps generated by various explainers for the VGG-16 model trained on the CUB-200-2011 dataset. Figure 7 highlights cases where the relevance maps produced by CLAM demonstrate high performance. For instances where CLAM is applied to LRP and CLRP, it is evident that regions irrelevant to the surrounding objects are eliminated. As observed in previous experiments, SGLRP inherently possesses the capability to remove surrounding areas, resulting in negligible changes when CLAM is applied. Examining the outcomes from explainers other than LRP, the gradient's result is noticeably noisy, resembling a simple visualization of activation. Guided_Backprop shows some noise reduction compared to the gradient method, although most of its results bear resemblance to edge detection. Grad-CAM effectively identifies areas pertaining to the main object; however, it requires resizing the relevance map generated from the last convolutional layer of the image classifier to the input size, which prevents obtaining precise pixel-level relevance maps and often includes surrounding noise. Excitation_Backprop successfully locates pixels related to the object but fails to eliminate peripheral noise.

Figure 8 shows instances where CLAM yields low performance. Upon observing CLAM's results in the figure, in many instances, areas pertinent to the main object are often eliminated. CLAM, designed to locate and emphasize regions related to the target class while relatively weakening oth-

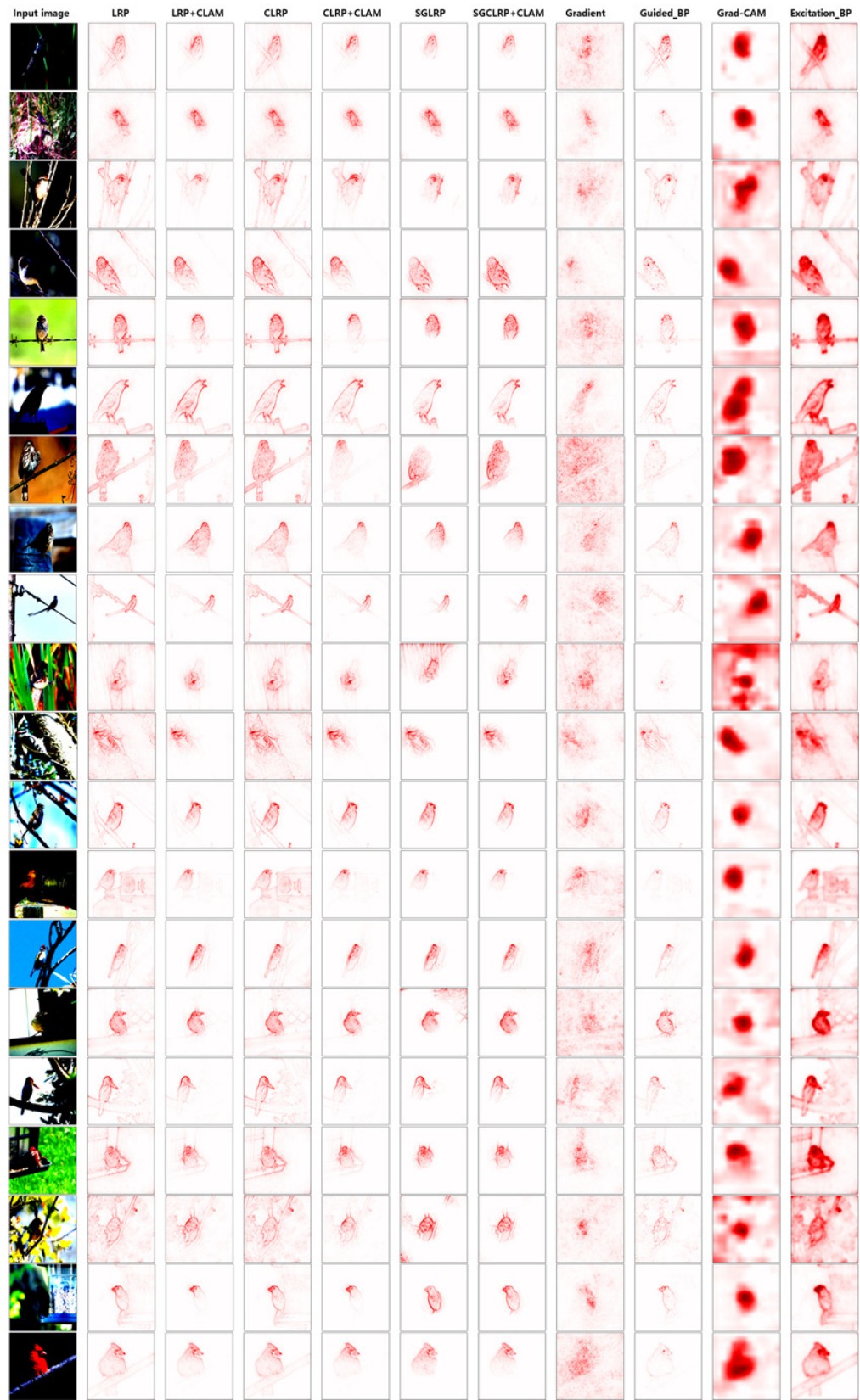

Figure 7: Results on generated relevance maps from various explainers for VGG-16 pre-trained on the CUB-200-2011 Dataset **(High CLAM Performance)**.

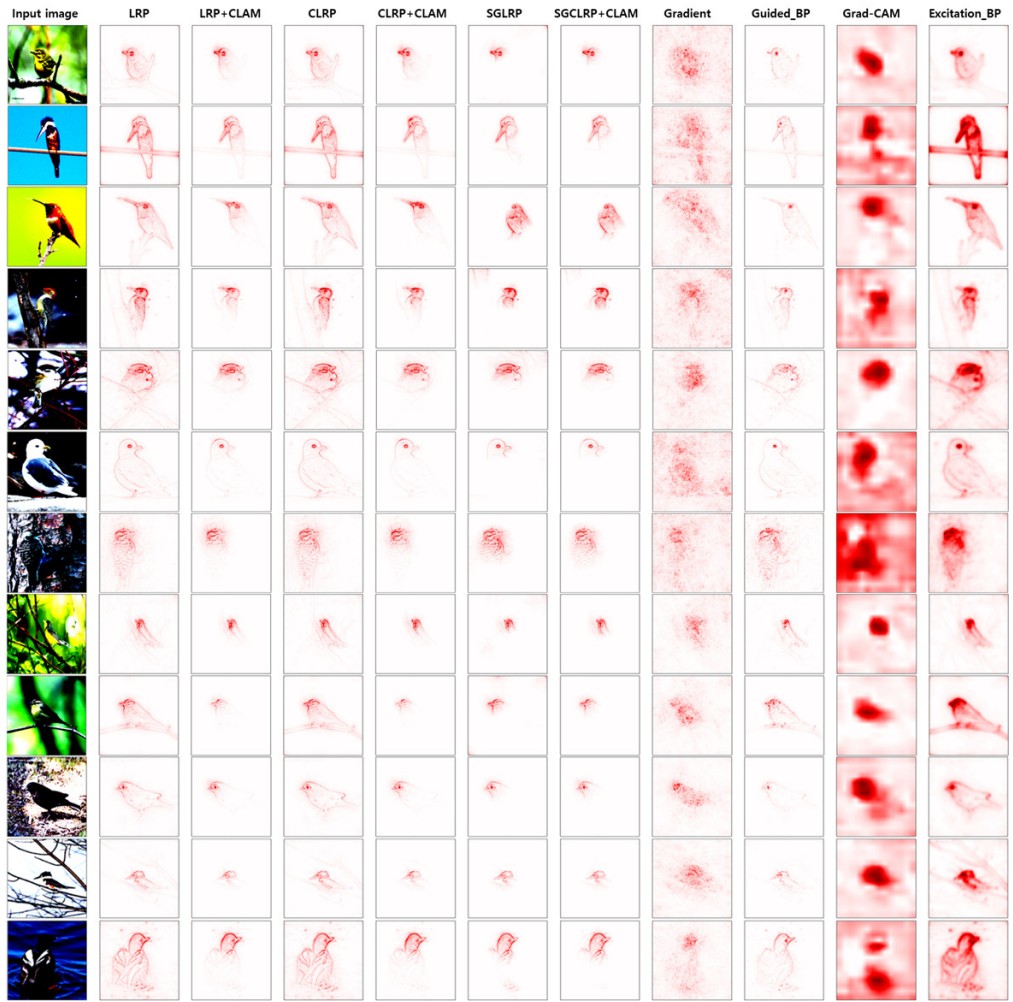

Figure 8: Results on generated relevance maps from various explainers for VGG-16 pre-trained on the CUB-200-2011 Dataset (**Low CLAM Performance**).

ers, may inadvertently yield detrimental effects, especially if the given image is overly simplistic. For instance, in the sixth row of Figure 8, the target object (bird) is centrally located with a simple background, resulting in well-generated relevance maps by LRP. When CLAM is applied to these results, it identifies the 'head' of the bird as the area most relevant to the current class, inadvertently weakening the relevance of the body in the final map. Since there are no definitive correct answers for these predictive relevance maps, neither case is inherently wrong. However, if we assume equal importance should be assigned across the entire object area, the result from CLAM cannot be considered superior to the original LRP. Similar observations, where the body of the object is eliminated, can be made in other cases, aligning with the lower average IoU results for LRP and CLAM as discussed in Appendix A.5.

Figures 9 and 10 illustrate the relevance map generation results of various explainers for the ResNet-50 model trained on the CUB-200-2011 dataset. Examining Figure 9 first, it is noticeable that applying LRP introduces considerable noise outside of the object area, owing to the incorporation of features from BatchNorm layers during the relevance map creation process in our experiment. Upon the application of CLAM, much of this noise is mitigated, leaving predominantly the regions related to the class-specific objects. Results from other explainers are consistent with those previously observed with the CUB dataset. Figure 10 provides examples with lower performance of CLAM similar to Figure 8. While employing CLAM effectively reduces surrounding noise, attempts to eliminate excessive noise occasionally lead to the removal of essential object areas, as evidenced

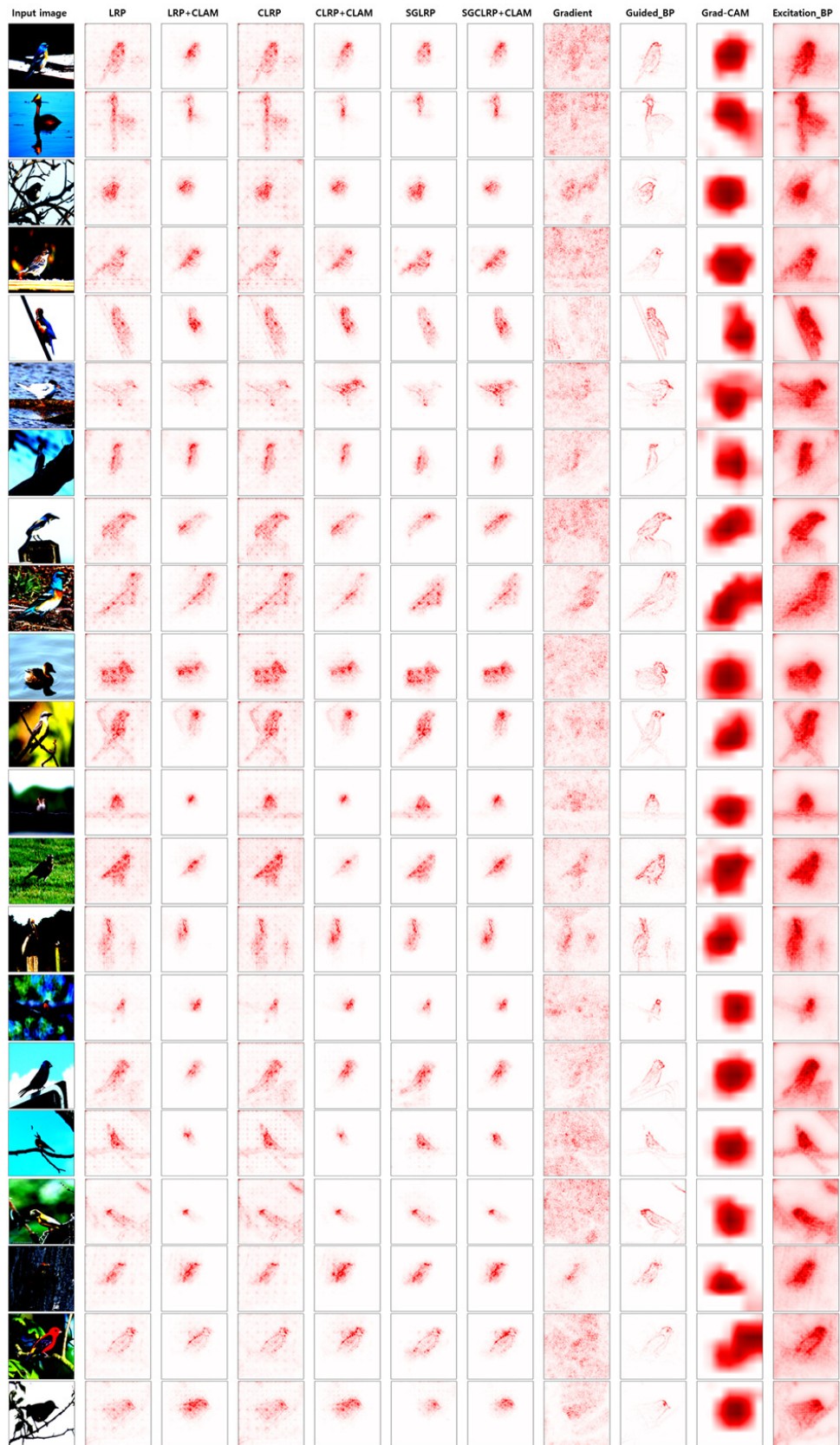

Figure 9: Results on generated relevance maps from various explainers for ResNet-50 pre-trained on the CUB-200-2011 Dataset (**High CLAM Performance**).

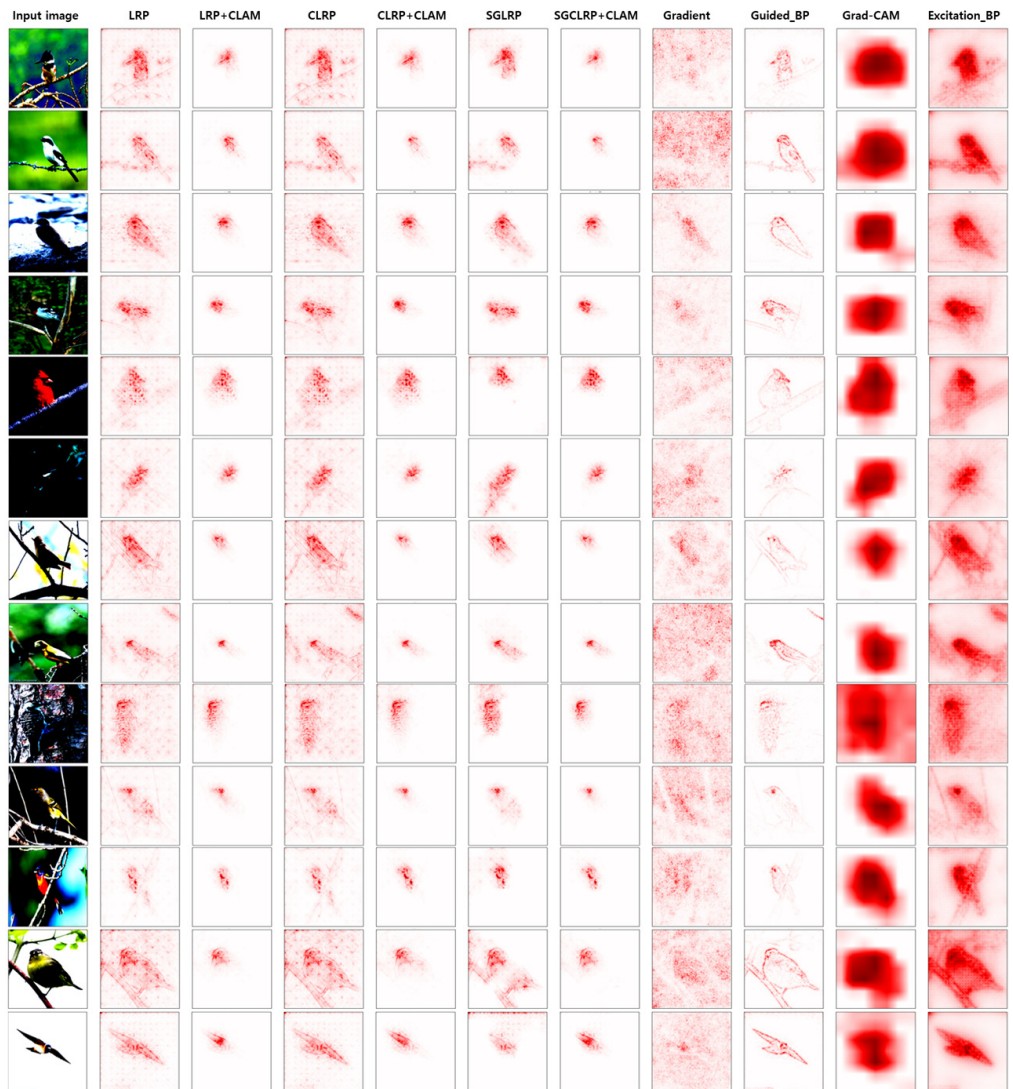

Figure 10: Results on generated relevance maps from various explainers for ResNet-50 pre-trained on the CUB-200-2011 Dataset **(Low CLAM Performance)**.

in several instances. This phenomenon is especially prominent in scenarios with simplistic backgrounds, like in the last row of Figure 10; areas within the object identified by the attributes become emphasized, consequently weakening the relevance values in other parts of the object.

Figures 11 and 12 depict the relevance map generation results by various explainers for VGG-16 and ResNet-50, respectively, both trained on the ImageNet dataset. Observing Figure 11, when comparing original LRP algorithms with those where CLAM is applied, the effects seem modest, slightly emphasizing certain regions within objects or mildly eliminating background noise, rather than significantly reducing noise as seen with the CUB-200-2011 dataset. Upon analysis, this difference is attributed to the configuration of the CLA-layers. For the CUB-200-2011 dataset, CLA-layers were connected to all convolutional layers of every image classifier. However, due to the substantial number of classes and corresponding large dataset size in ImageNet, establishing CLA-layers for all convolutional layers would require significant training time. Consequently, we selected five primary convolutional layers for which CLA-layers were configured and trained. While this approach considerably reduced the overall training time, the limited number of CLA-layers also diminished the efficacy of relevance map adjustments by CLAM. This experience indicates the necessity for strate-

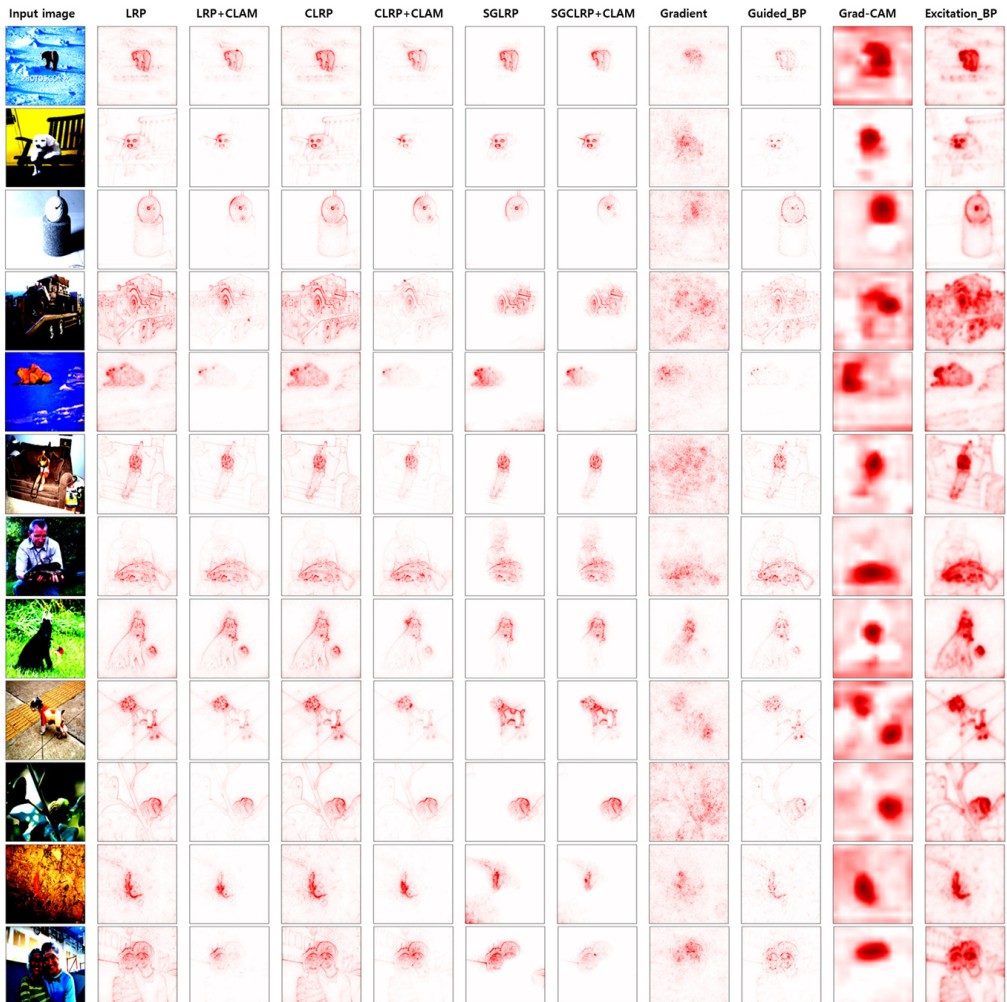

Figure 11: Results on generated relevance maps from various explainers for VGG-16 pre-trained on the ImageNet2012 Dataset.

gic configuration of CLAM's CLA-layers depending on the architecture of the target image classifier and the size of the dataset. Further discussion on this matter can be found in Appendix A.6.

As evidenced by various quantitative evaluations previously discussed, implementing CLAM on ResNet-50 is more challenging than on VGG-16 due to its deeper and more complex structure. Therefore, we devised two versions of CLAM for application on ResNet-50. Case 1 involves training with the final features of all Bottleneck blocks and the features from the $3 \times 3$ convolutional layers within the Bottleneck, while Case 2 utilizes only the last features of the Bottleneck blocks. In Case 1, CLAM contains an excessive number of CLA-layers, leading to frequent over-adjustments of the relevance map during backpropagation following the LRP algorithm, often eliminating correct regions. Figure 13 demonstrates this outcome, with many instances where the application of CLAM inadvertently removes the entire object area. Although Case 2 fares better than Case 1, it omits much of ResNet-50's residual feature information, providing only marginal noise reduction without substantial improvement over the original LRP algorithm, as seen in Figure 12. Through these results, we acknowledge the limitations of CLAM and aim to explore effective applications of CLAM on complex structures in future work.

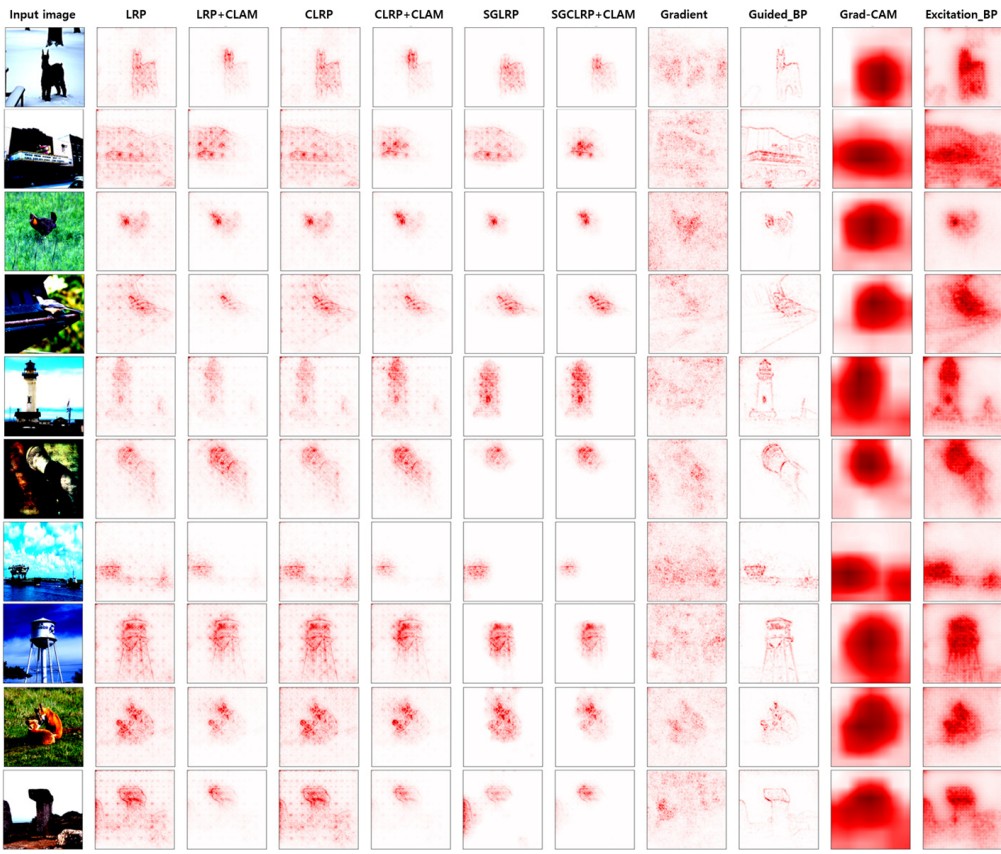

Figure 12: Results on generated relevance maps from various explainers for ResNet-50 pre-trained on the ImageNet2012 Dataset.

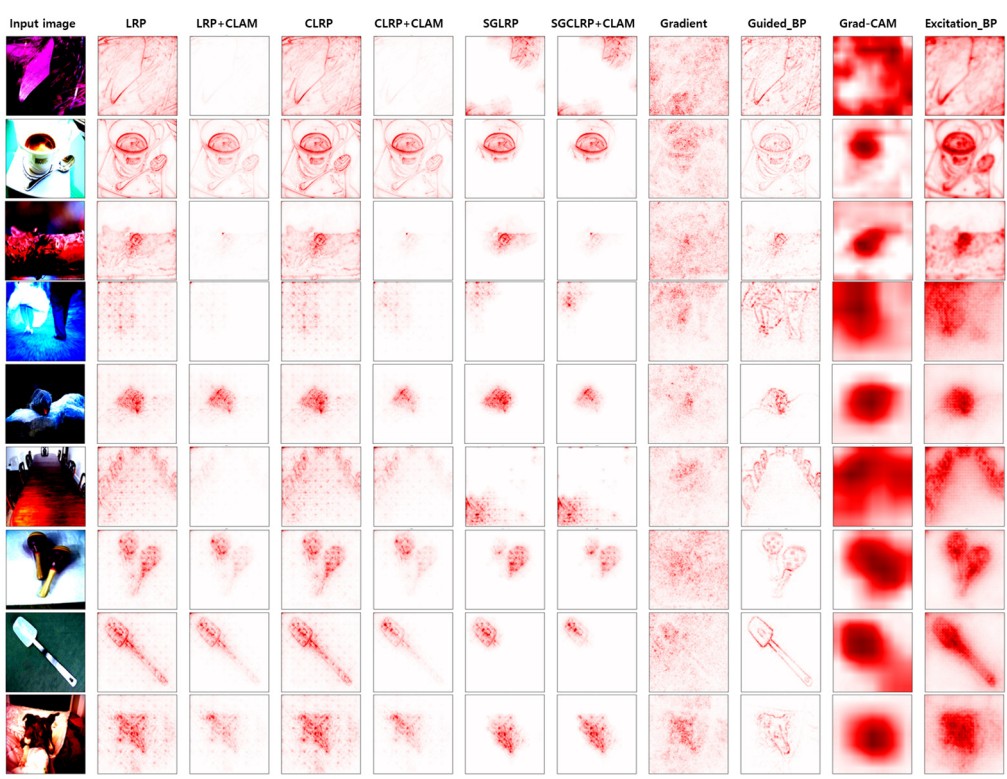

Figure 13: Results on generated relevance maps from various explainers for ResNet-50 pre-trained on the ImageNet2012 Dataset (**Low CLAM Performance**).

### A.4 ADDITIONAL EXAMPLE-BASED EXPLANATIONS AND ANALYSIS

As indicated in Section 4.2.2, CLAM was capable of providing example images from the training dataset related to the attribute regions reflected in the map when generating a relevance map from an input image via learned attributes. These example images offer additional insights into which images from the actual training dataset influenced the regions reflected in the relevance map. Figures 14, 15, and 16 illustrate examples of images from the training dataset and their corresponding bounding

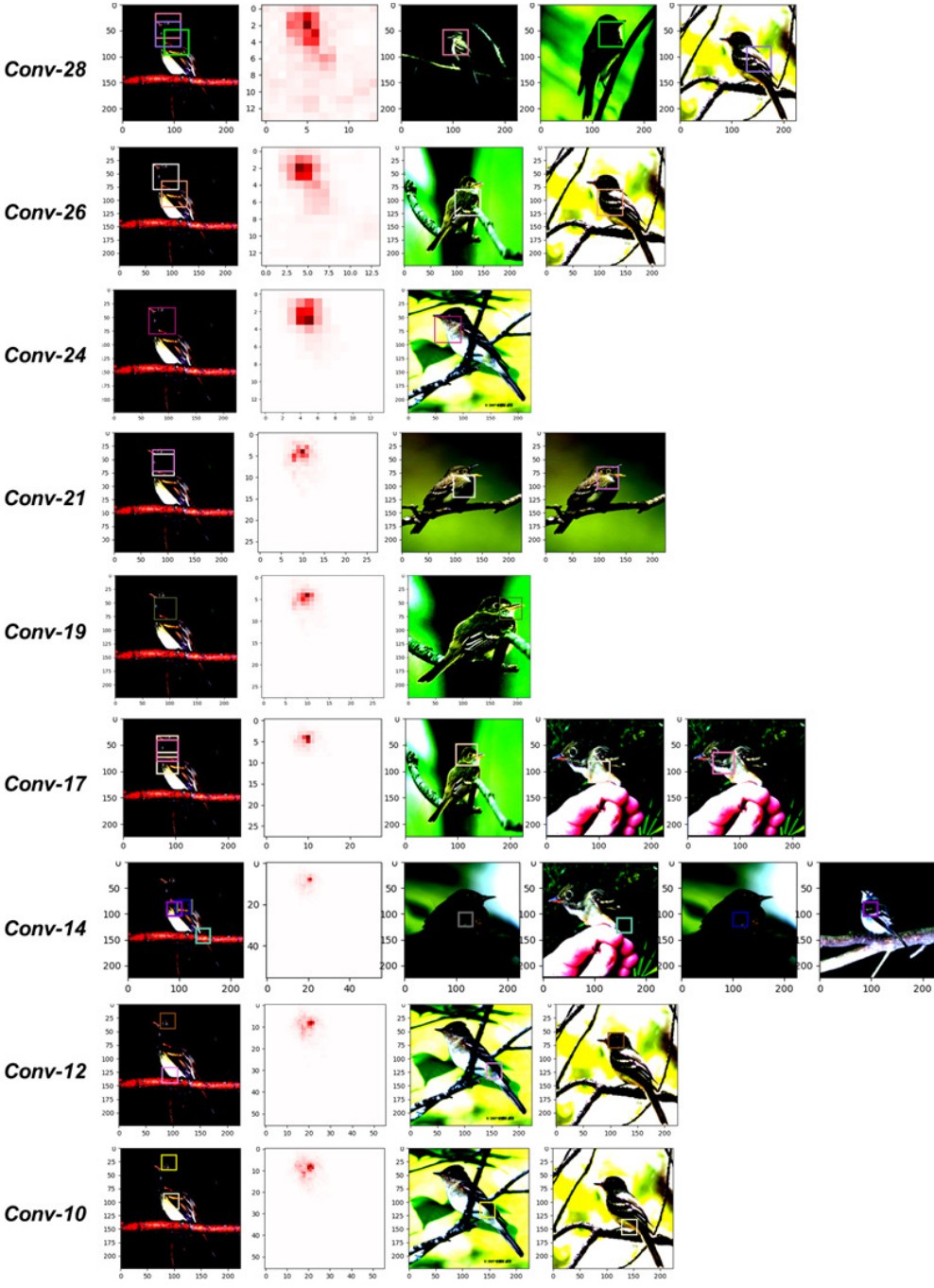

Figure 14: Example 1. Images related to attributes regions reflected in relevance map for each CLA-layer

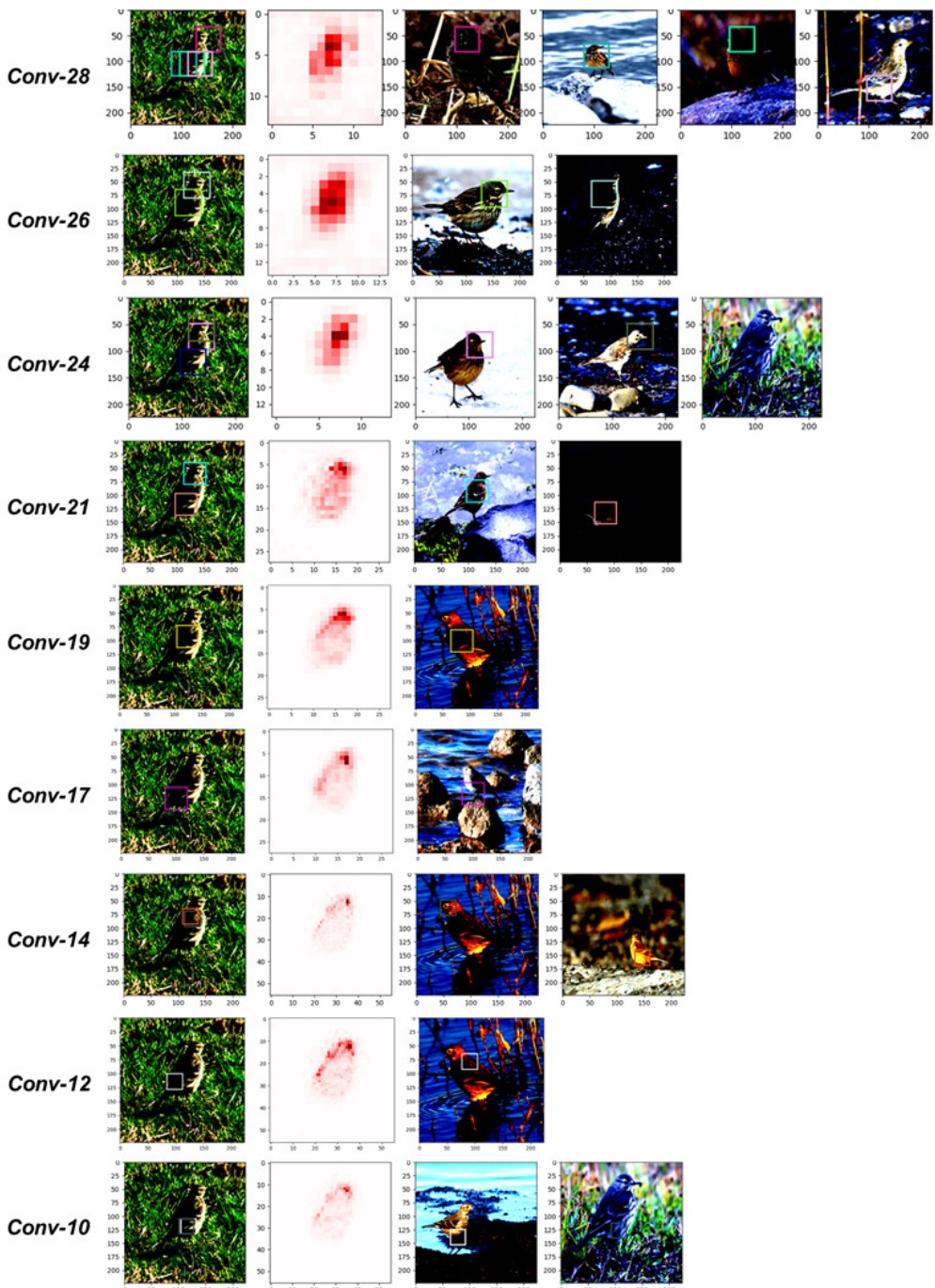

Figure 15: Example 2. Images related to attributes regions reflected in relevance map for each CLA-layer.

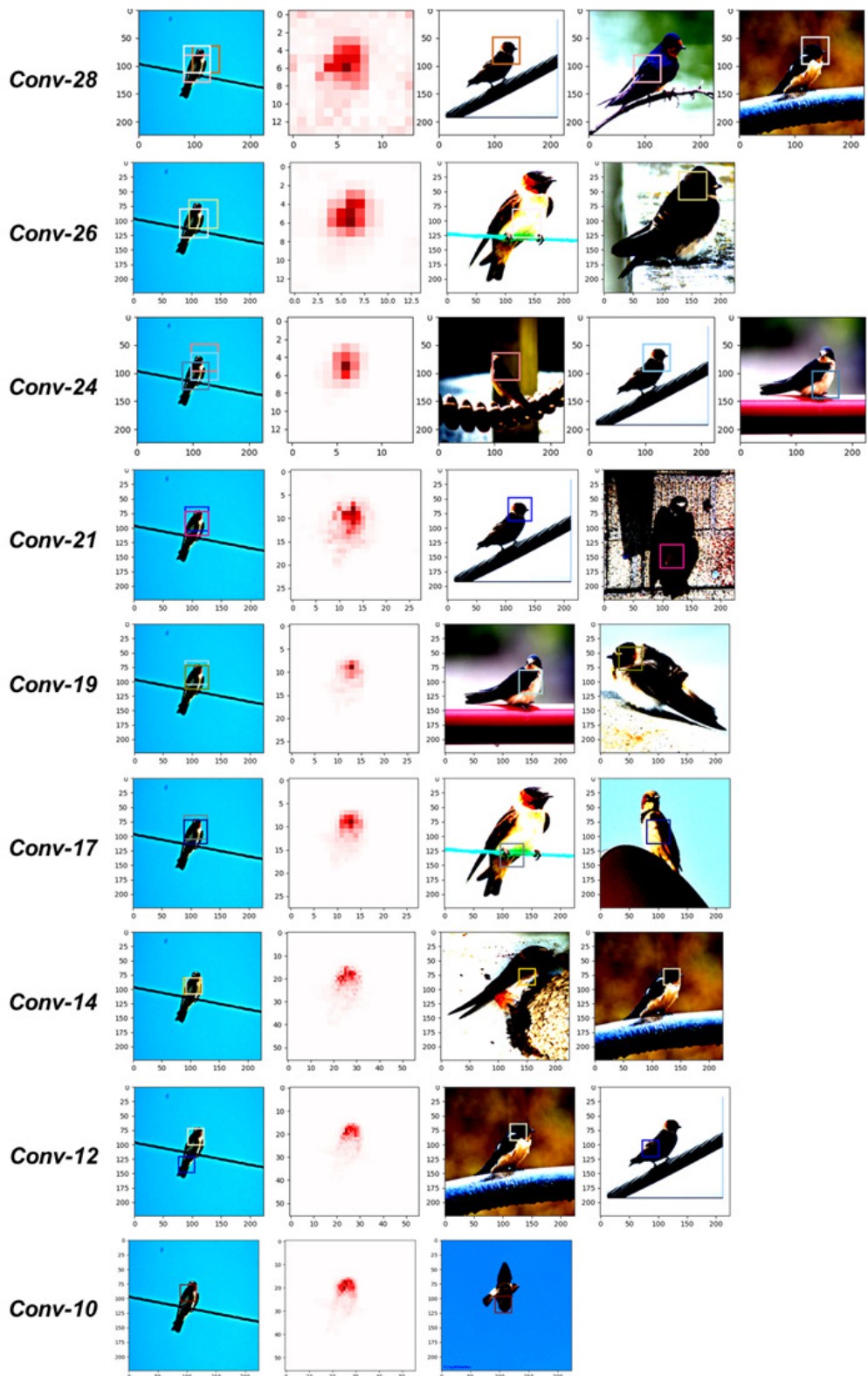

Figure 16: Example 3. Images related to attributes regions reflected in relevance map for each CLA-layer.

boxes that are related to the attribute regions reflected in the relevance map, with each layer being displayed separately. The first column lists the names of convolutional layers where the CLA-layer is applied. The second column shows the important regions at each layer for the input image and the target class determined by the attribute. The third column presents the relevance map at each layer, where the values of the regions selected by the attribute are enhanced through masking. Subsequent columns provide the training dataset image (and its region) that is most similar to each region selected by the attribute. It notes that the CLA-layers closer to the input layer have minor influence from the attributes and are, therefore, omitted.

Upon examining each figure, it can be observed that areas primarily associated with the target class tend to be mapped together. For instance, in Figure 14, from 'Conv-28' to 'Conv-17', the regions selected by the attributes of each layer predominantly pinpoint the 'head' of the bird as the most crucial area for the class. Correspondingly, images from the training dataset also predominantly indicate the bird's head. Moreover, the relevance values in the relevance map for the head area are also enhanced. Considering 'Conv-14', one of the attributes selects the bird's tail portion (indicated by a cyan box) from the input image. Images from the training dataset related to this attribute also pinpoint the tail portion of a bird consistently. However, even when the area of this attribute is reflected in the relevance map, it is not emphasized in the actual relevance map since its values are significantly lower than those of the head area previously enhanced. From the model's perspective, when information is aggregated layer-wise for the target class, the head region can be identified as significantly influential to the class overall. Such results are beneficial in the context of highlighting areas that contribute substantially to model predictions, as illustrated by the example. Nonetheless, this might not be advantageous when it comes to evaluations involving Intersection over Union (IoU) for segmentation or when there's a need to locate the entire object area. A straightforward solution to this issue is to adjust the 'strength' with which attribute regions are reflected in the relevance map through parameterization mentioned in Appendix A.1. In future developments and enhancements of CLAM, various considerations and factors need to be taken into account.

Figures 17 and 18 present examples where the accuracy of attributes is suboptimal. Firstly, Figure 17 illustrates a problem arising from the difference in object sizes between the test image and the connected images from the training dataset. Upon observing the input image in the figure, it's evident that the target object area is relatively small. Consequently, the regions selected by the attribute usually encompass more than half of the object. However, some images from the training dataset identified by the attribute contain substantially larger objects. For example, in 'Conv-17' of Figure 17, the attributes identified an area from the input image that corresponds to the entire body of the bird, but for related training set images providing images pointing to both parts of the bird's body and beak. While this isn't incorrect at a glance, a more detailed examination reveals a mismatch in the levels of the areas within the bounding boxes. This issue stems from the fact that the patch sizes of each CLA-layer are fixed. Such findings could be pivotal in future developments and improvements of CLAM. Figure 18 depicts instances where the mapping between the regions found by the attributes is inaccurate. For instance, in 'Conv-19', while the blue box points to the bird's head in the input image, the image from the training dataset indicates the bird's leg. This problem occurs due to the imperfect performance of the trained attributes themselves, highlighting through this example the need for evaluation of attributes themselves.

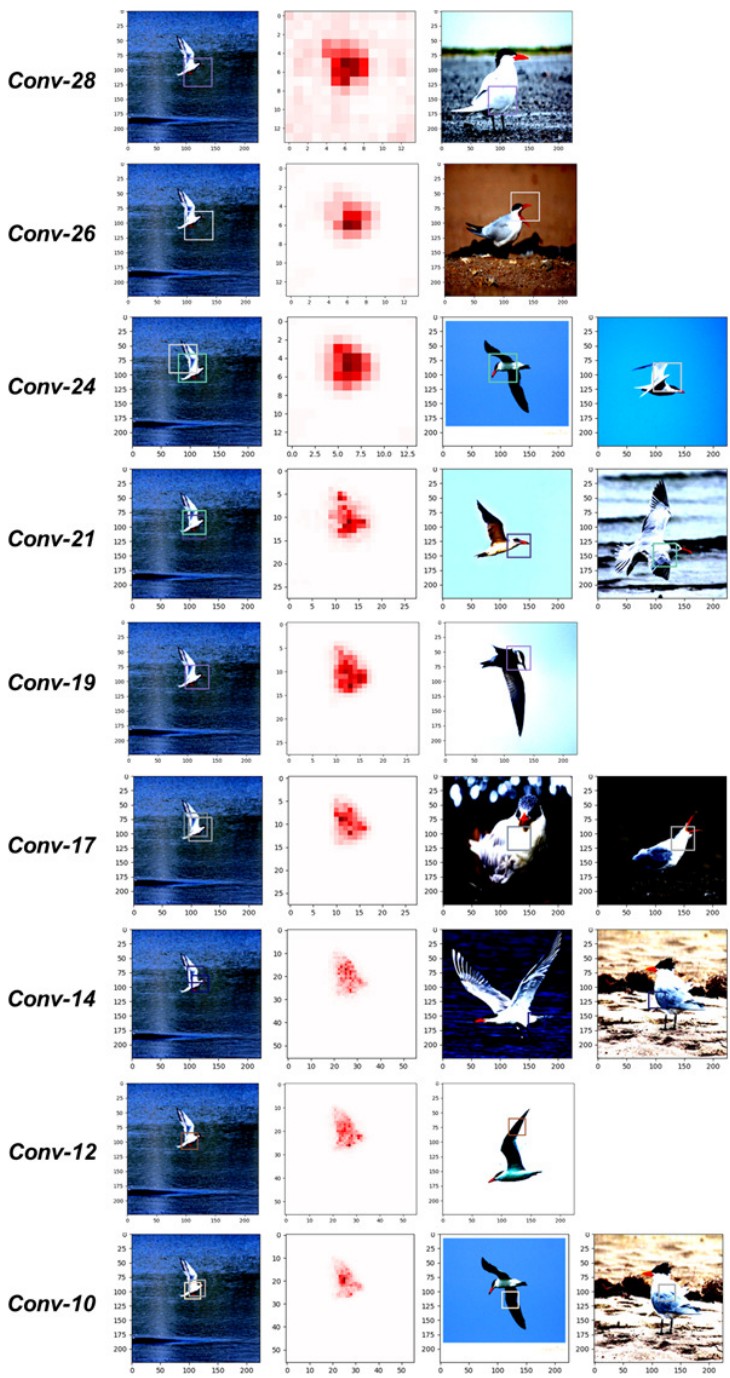

Figure 17: Example 4. Images related to attributes regions reflected in relevance map for each CLA-layer **(Incorrect case)**

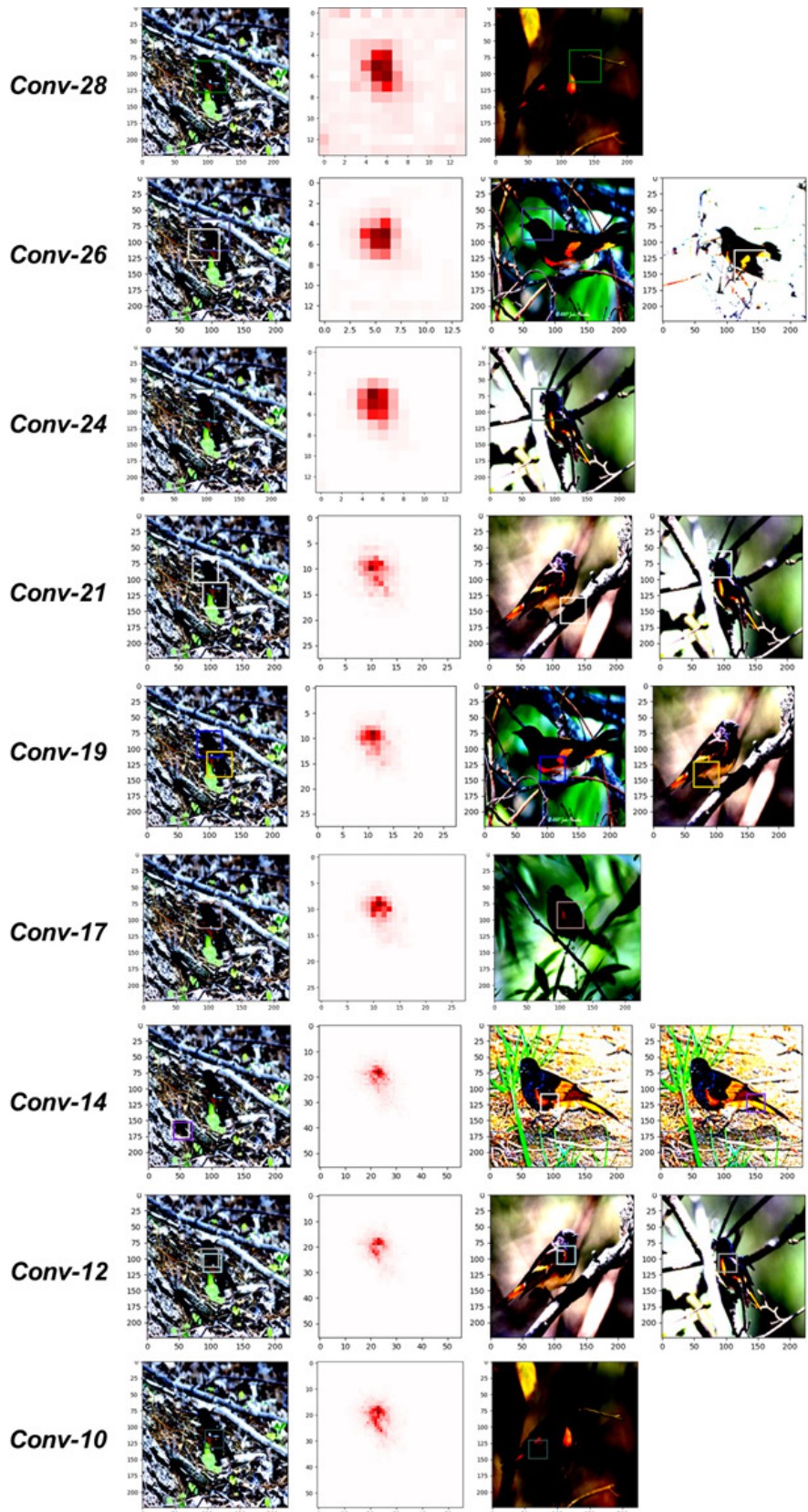

Figure 18: Example 5. Images related to attributes regions reflected in relevance map for each CLA-layer **(Incorrect case)**

## A.5 Evaluation Through average IoU for Segmentation

In additional experiment, using the segmentation masks provided in the CUB-200-2011 datase, the average IoU performance of the relevance maps generated by each explainer is measured and compared. Unlike the IoU for bounding boxes, segmentation allows for area evaluation at the pixel level, as it utilizes masks labeled at this granularity. For IoU calculations, we employ the localization method executed in (Lee et al., 2021). To perform segmentation on the relevance maps generated by the explainer, masks are created, consisting of values exceeding average + 1*standard deviation. The IoU is then computed using these created masks and the ground truth masks provided in the dataset.

Table 4 presents the average IoU values for relevance maps generated by explainers, each associated with image classifiers trained on respective datasets. Firstly, it is evident that both LRP and CLRP exhibit improved performance when integrated with CLAM, as the latter helps reduce noise from areas outside the object, as previously verified. Conversely, SGLRP shows a decrease in performance. This decline is attributed to CLAM inadvertently eliminating areas within the object while reducing noise on already denoised relevance maps produced by SGLRP. When comparing the performance of CLAM with other explainers, it is observed that CLAM generally underperforms relative to Grad-CAM and Excitation_Backprop. This discrepancy is because, for higher IoU performance, it is imperative not only to eliminate noise but also to preserve the object areas intact. While creating masks for relevance maps, pixel values corresponding to noise get removed, thereby diminishing the noise reduction effect of CLAM. Furthermore, since CLAM primarily focuses on noise reduction, the inadvertent removal or alteration of some object areas significantly compromises its IoU performance, resulting in a lower average performance. As can be seen in Figure 3, while CLAM applications yield minimal noise, there are instances where parts of the object area are slightly erased. Addressing this issue is a challenge that future iterations of CLAM must tackle, especially from a segmentation perspective.

Table 4: Results on average IoU score with segmentation mask of CUB-200-2011.

|  | LRP | LRP+C | CLRP | CLRP+C | SGLRP | SGLRP+C |
|---|---|---|---|---|---|---|
| VGG-16 | .2244 | .3693 | .2693 | .3541 | .3088 | 2256 |
| ResNet-50 | .2762 | .3541 | .2630 | .3885 | .3540 | 2008 |

|  | LA. | Deconv. | Gradient | Guided-GP. | Grad-CAM | Ex-BP. |
|---|---|---|---|---|---|---|
| VGG-16 | .1343 | .1358 | .2820 | .2766 | .4240 | **.4751** |
| ResNet-50 | .0992 | .1649 | .1685 | .2834 | .4180 | **.4531** |

## A.6 Limitations And Future Plans

CLAM was developed with an aim to amalgamate the advantages of both traditional post-hoc explanation methods and the transparent design approach, thereby enhancing the depth and performance of the explanations provided. Particularly, while post-hoc explanation approaches boast the benefit of applicability across various models through predetermined algorithms, they fall short in understanding and generating explanations for the target models themselves. In response to this limitation, the inception of CLAM was characterized by designing a model capable of statistically analyzing and modeling the abundant feature and relevance information derived from numerous data inputs fed into an already trained model. To achieve this, CLA-layers were crafted, and through the iterative learning of abundant features and relevance maps across multiple CLA-layers, it was possible to acquire attribute information specific to each class and layer. This process not only enhanced the performance of the generated relevance maps but also facilitated the provision of additional explanations.

However, through various experiments, several limitations were identified. Firstly, there is difficulty in generalizing for different models. When comparing VGG-16 and ResNet-50, which were utilized in this study, substantial differences were observed even when both were trained with the same dataset and exhibited similar performance. Notably, due to its multiple residual connections, block-

like structures, and deep neural network architecture, ResNet-50 necessitates many CLA-layers, leading experimentally to a decline in the performance of relevance maps. To address these issues, a simple solution would be adjusting various hyperparameters. For example, the number of CLA-layers can be set appropriately depending on factors such as the depth of the model, the number of convolutional layers, the size of the model's features, the size of the utilized dataset, and the number of classes. Alternatively, after the training of CLA-layers, during the test phase, the 'strength' parameter can be separately configured to adjust the reflection of attributes in the relevance map. Moreover, it's unreasonable to assume that all features of the convolutional layer equally affect predictions. For instance, in the case of Grad-CAM, utilizing only the last convolutional layer of the model to create relevance maps still yields excellent performance. This suggests that as features are closer to the output layer, their importance increases; therefore, when constructing CLA-layers, it may be beneficial to allocate more significance to those closer to the output layer. In this study, the CLA-layers near the input layer are indeed set with lower strength values when adjusting the relevance maps.

Beyond immediate performance improvement, considerations regarding the structure of CLA-layers and the loss function for training CLA-layers appear necessary for the broader application of CLAM across various models. Particularly, as models based on Transformers (Vaswani et al., 2017; Dosovitskiy et al., 2020; Han et al., 2021; Chang et al., 2022), which possess more complex structures than traditional CNN-based deep learning models, are being extensively utilized across diverse fields, there is a pressing need for progressive advancements to apply CLAM effectively to these models. There are instances where LRP methods are applied to Transformer-based models (Abnar & Zuidema, 2020; Chefer et al., 2021; Ali et al., 2022; Naseer et al., 2021; Chefer et al., 2021), necessitating further developments for the implementation of CLAM on such architectures. The final goal of CLAM is to analyze the independent space that generates features, irrespective of the model structure, and to collaborate with existing explainers capable of generating explanations, thereby enhancing explanatory power. Future research will be conducted aligned with this objective. Another identified limitation is the absence of an evaluation method specifically for CLAM itself. Since there are no predefined correct answers for the sub-feature areas extracted by the attributes learned at each CLA-layer in CLAM, there is not a direct way to assess the accuracy level of the attributes. In light of this, the experiments conducted in this study proceeded by reflecting the attributes in the relevance map and evaluating the relevance map itself. However, to generalize CLAM further and apply it to various tasks, there is a need to explore and develop evaluation methods specifically for CLAM. Future research is planned to address this need and advance this aspect of the framework.

