# OpenReview forum: "CLAM: Class-wise Layer-wise Attribute Model for Explaining Neural Networks"
_ICLR.cc/2024/Conference — ICLR 2024 Conference Withdrawn Submission_

### Official Review · Reviewer_wRSa · 2023-10-20

**Soundness:** 2 fair
**Presentation:** 1 poor
**Contribution:** 2 fair
**Rating:** 3
**Confidence:** 4

**Summary:**

The paper proposes a method to generate explanations for neural networks by proposing a method called CLAM. CLAM is an extension of LRP methods such that it could be put on top to improve the performance of LRP approaches. For this, CLAM adds trainable parameters to the neural network such that LRP maps are guided by attributes (or prototypes) that are learned from data by the forward-forward algorithm. The generated explanations are quantitatively and qualitatively evaluated using CUB.

**Strengths:**

1. The proposed method is evaluated with respect to qualitative and quantitative measures. (quality)
2. The proposed method sounds promising as it improves LRP (a well-known technique for the explanation of neural networks) by contextualizing attributes from the dataset. (originality+ significance)
3. The authors present several results to show convincingly the performance of the proposed method. (quality)

**Weaknesses:**

Please see for further information Questions where I will explain these points in more detail.

1. The paper is difficult to understand and not well-written. Especially the math (algorithm structure) is difficult to understand. (clarity)
2. The experiments should be improved. Right now, it is not clear if, for instance, the example explanations are generated on correct or incorrect predictions.
3. The results should also be discussed alongside the dimension of computational and time complexity.

**Questions:**

I recommend that the authors read these questions carefully to improve the quality of the submission. If these concerns/questions can be clarified, I will reassess my scoring of the paper.

1. Could the authors improve the understandability of Section 3? Right now, it is really difficult to read. I think the description would really benefit from the presentation of pseudo-code. Moreover, I recommend to revise the notations used. Using bold symbol for dimensions is rather untypical and confusing as bold symbols are usually used for matrices or tensors. Additionally, the notations between the appendix and the main paper should be aligned. Right now, they use different notations; the appendix uses 'sz_attrs' and so on. Furthermore, in Section 3.2 there are several repetitions of sentences. This should be fixed.
2. Section 4.2.1: The first and the second paragraph are almost identical. Could the authors fix this?
3. Is there a non-linear activation between the two 1x1 conv layers?
4. Figure 2: How is the distance computed if P>A, and why is it strictly greater than?
5. What size has D?
6. Maybe I missed this in the paper: Why is the dimensionality of S "+1"?
7. Section 3.3, ICP: How does such a matrix pattern look?
8. Please note that Equation 1 consists of at least two disconnected equations, which should be fixed. Moreover, where is the index n in the lower right equation? Where is the index i in the upper equation (it is only an AvgPool without any dependence on i)? What is the meaning of ",:" in the equations?
9. Equation 2: What is $||\cdot||$?
10. What is "Distance" in Equation 3?
11. Is the BCL loss or the Softplus function used as a loss function (see Section 3.3.)?
12. If the authors want to compare the performance of explanations with respect to the localization properties, I recommend to use the following benchmark dataset: Leila Arras and Ahmed Osman and Wojciech Samek: CLEVR-XAI: A benchmark dataset for the ground truth evaluation of neural network explanations. Is there a reason why this dataset cannot be used?
13. Section 4.2.1: How do the authors know that the background is always attributed to noise? Maybe the method highlights context information that is used. Same applies to later statements that are similar such as "filters out areas irrelevant to the target."
14. Could the authors elaborate on how to interpret/explain the decision process based on the derived maps? What changes if the map for the second highest prediction is considered? Does the map explain why the network hasn't decided for this class? In general, I recommend that the authors carefully refine their evaluation. An XAI method should help developers and users to understand the decision process of a network. This should go beyond presenting images that look reasonable (or satisfy the personal confirmation bias). The authors should challenge their XAI technique. For example, explain (using the maps) why a network makes certain errors or why it can be fooled by adversarial examples. Only presenting the images doesn't help in understanding the decision process. Since the authors haven't reported which predictions have been used, I guess they always present samples of correct predictions (note that the class label should be mentioned).
15. What does removing mean in Section 4.2.1? Setting the values to 0?

Some notes/comments to the authors, which I have not considered for the scoring:

1. Please have you paper proofread by an English native speaker.
2. Abstract: There is some weird character in "Class-wise Layer-wise Attribute M¡model (CLAM)."
3. Please use either $R^{(l)}_{2d}$ or $R^{(l)}{2d}$.

**Details Of Ethics Concerns:**

None.

---

### Official Review · Reviewer_vhXF · 2023-10-30

**Soundness:** 2 fair
**Presentation:** 2 fair
**Contribution:** 2 fair
**Rating:** 3
**Confidence:** 4

**Summary:**

In this paper, the author present CLAM to study the interpretability of models. To learn attributes at each class and layer level, CLAM is designed to use with a pre-trained image classification model and an existing interpretable algorithm to learn class-wise layer-wise attributes from the model features. User studies were conducted to show the explainability ability of this method.

**Strengths:**

1. The authors argue they enhance the explainability of image classifiers, which may be meaningful in the relative fields.
2. The paper is well-organized and easy to follow.

**Weaknesses:**

1. The authors is built with certain modifications specific to approximate the target model, which compromises the technical novelty of this work. Besides, is the method able to explain ViT models?

2. The author did not discuss the training time of CLAM layers, and th selection of parameters such as number of attributes for different layers & datasets are also missing, which should influence the final performance.

3. The comparison methods in Tables 1 & Figure 3 are vanilla so that they not quite convincing. Maybe more stronger methods need to be considered. In addition, the baseline methods, i.e., LRP and Grad-CAM, used in Tables 1 & Figure 3 are proposed in 2015 and 2016 respectively. More recent methods should be considered as baselines, for example, “On locality of local explanation models, NeurIPS 2021”, “Craft: Concept recursive activation factorization for explainability, CVPR 2023”, “RKHS-SHAP: Shapley values for kernel methods, NeurIPS 2022”.

4. The quantitative results are not satisfying. The evaluation result reported in Table 1 only show minor improvement compared to Grad-CAM, and most of recent methods are missing in the table.

**Questions:**

The author only integrated CLAM with LRP-based methods, which is quite confusing to me. Have the author tried to integrate with other methods?

---

### Official Review · Reviewer_fMxS · 2023-10-31

**Soundness:** 2 fair
**Presentation:** 1 poor
**Contribution:** 2 fair
**Rating:** 3
**Confidence:** 4

**Summary:**

This paper proposes a method that can be used to generate explanations of image classifiers. I can't fully understand the exact method because the paper uses a number of undefined terms and concepts to explain the methods, but it tries to generate a relevance map from model prediction. Evaluation was performed using public image datasets where the proposed method was shown to sometimes improve the accuracy by around 1% and sometimes degrades the accuracy.

**Strengths:**

The topic is relevant and important.

The paper performed both qualitative and quantitative analysis.

**Weaknesses:**

I feel this paper needs to explain itself better given the the paper is about explainability. I don't understand many sentences and paragraphs written in this paper and feel this is largely because the paper lacks a structure to clearly articulate its core development, the writing is unclear, and many of the key terms are never defined.

For example, I can't understand anything from this text: "One approach generates contribution maps (or relevance maps), which use visualized internal model values to shed light on predictions. This method, however, is more about reverse-analyzing the model’s decision-making rather than understanding the model itself, providing a ‘visualization’ rather than an ‘explanation’ for predictions"

"Contribution maps" (or relevance maps) was never defined in the paper, but this uses "visualized internal model values". What are the internal model values? Activation, gradients, saliency, etc. It must be something. This is about "more about reverse-analyzing the model’s decision-making rather than understanding the model itself" -- then what do the authors mean by understanding the model itself other than analyzing the model's outputs? This is "visualization" not an "explanation". How do the authors define "explanation"? As far as I could tell, this paper defines "relevance map/visualization" = "explanation".

Another example is "explainers". I couldn't find the definition of this term while this term appears 51 times in the paper. I can only assume that this is something that gives us a relevance map.

Another example is "attribute". "Attributes fundamentally extract sub-feature areas related to specific classes using features". What is "sub-feature?" Attributes extract some areas, using features -- what features? Attributes are typically defined as semantics, i.e. word descriptions. Are there any semantics assigned for these attributes?

Sec 3.2 suddenly starts defining modules and their dimensions, but again, it is informal and I can't understand any of this without having a clear high level overview of the method.

The quantitative results show the proposed method doesn't quite improve the accuracy. The gain is at most 1.5% but it often even hurts the accuracy. I'm not convinced that this method is practically useful given the additional complexity added to the method.

**Questions:**

What is the target audience? Explainability papers should be easy to read by general audience, but even experts in this topic would have difficulty understanding the exact method.

---

### Official Review · Reviewer_U1pQ · 2023-10-31

**Soundness:** 2 fair
**Presentation:** 2 fair
**Contribution:** 2 fair
**Rating:** 3
**Confidence:** 3

**Summary:**

The paper addresses deep model explanation task with a method called Class-wise Layer-wise Attribute Model (CLAM), which is designed to work in conjunction with a pre-trained image classification model and an existing interpretable algorithm. When generating a relevance map for new input images, CLAM leverages the learned attribute information to enhance the areas related to the target class, leading to improved accuracy. The main processes of CLAM include the Forward-Forward algorithm and the Intrinsic Class Pattern approach.

**Strengths:**

1.The proposed CLAM usually improves existing approaches (LRP, CLRP, etc.) largely.

2. The CLAM seems to capture more accurate relevance maps.

3. Integrating the Forward-Forward algorithm and the Intrinsic Class Pattern algorithm is a good idea for class-wise attribute model learning.

**Weaknesses:**

1.There is vital problem of the written--two duplicated paragraphs occur in the bottom of the 3rd page!

2.The L2-Distance in Eq.(3) is weird.

3.From Table 1 and Table 2, the CLAM module added on existing methods usually leads to performance degradation when ResNet50 is used.

4.In Table2, some boldface numbers are not the best performance.

5.There is no explanation of what kinds of attributes have been learned.

**Questions:**

Please see the weaknesses.